https://doi.org/10.1038/s41467-020-14313-0　　**OPEN**

# Histidine kinase MHZ1/OsHK1 interacts with ethylene receptors to regulate root growth in rice

He Zhao [1,2,5], Kai-Xuan Duan[1,2,5], Biao Ma[3,5]*, Cui-Cui Yin[1], Yang Hu[1,2], Jian-Jun Tao[1], Yi-Hua Huang[1,2], Wu-Qiang Cao[1,2], Hui Chen[1,2], Chao Yang[1], Zhi-Guo Zhang[4], Si-Jie He[1], Wan-Ke Zhang[1], Xiang-Yuan Wan[3], Tie-Gang Lu[4], Shou-Yi Chen[1]* & Jin-Song Zhang[1,2]*

Ethylene plays essential roles during adaptive responses to water-saturating environments in rice, but knowledge of its signaling mechanism remains limited. Here, through an analysis of a rice ethylene-response mutant *mhz1*, we show that *MHZ1* positively modulates root ethylene responses. *MHZ1* encodes the rice histidine kinase OsHK1. MHZ1/OsHK1 is autophosphorylated at a conserved histidine residue and can transfer the phosphoryl signal to the response regulator OsRR21 via the phosphotransfer proteins OsAHP1/2. This phosphorelay pathway is required for root ethylene responses. Ethylene receptor OsERS2, via its GAF domain, physically interacts with MHZ1/OsHK1 and inhibits its kinase activity. Genetic analyses suggest that MHZ1/OsHK1 acts at the level of ethylene perception and works together with the OsEIN2-mediated pathway to regulate root growth. Our results suggest that MHZ1/OsHK1 mediates the ethylene response partially independently of OsEIN2, and is directly inhibited by ethylene receptors, thus revealing mechanistic details of ethylene signaling for root growth regulation.

---

[1] State Key Lab of Plant Genomics, Institute of Genetics and Developmental Biology, INASEED, Chinese Academy of Sciences, 100101 Beijing, China. [2] College of Advanced Agricultural Sciences, University of Chinese Academy of Sciences, 100049 Beijing, China. [3] Biology and Agriculture Research Center, School of Chemistry and Biological Engineering, University of Science and Technology Beijing, 100024 Beijing, China. [4] Biotechnology Research Institute/National Key Facility for Genetic Resources and Gene Improvement, Chinese Academy of Agricultural Sciences, 100081 Beijing, China. [5] These authors contributed equally: He Zhao, Kai-Xuan Duan, Biao Ma. *email: mabiao@ustb.edu.cn; sychen@genetics.ac.cn; jszhang@genetics.ac.cn

Ethylene plays essential roles in plant growth and development. In the model plant *Arabidopsis*, a linear ethylene signaling pathway has been established based on the characterization of a series of triple response mutants[1–4]. The pathway contains a family of endoplasmic reticulum (ER) membrane-bound ethylene receptors, a Ser/Thr kinase CTR1 (constitutive triple response 1), a central ER membrane protein EIN2 and transcription factors EIN3/EIL1 and ERF1[5–13].

In the absence of ethylene, the ethylene receptors are in active state and CTR1 is likely activated to phosphorylate the C-terminal domain of EIN2 to repress ethylene response. In the presence of ethylene, the C-terminus of EIN2 is cleaved and translocated to nucleus for activation of downstream EIN3/EIL1 transcriptional cascade and then ethylene response[14–16]. EIN2 and EIN3/EIL1 are regulated by proteasomal degradation[17–20]. Recently, the EIN2 C-terminal domain is also found to be targeted to the cytoplasmic processing-body (P-body) for the translational regulation of *EBF1/2*[21,22]. EIN2-mediated histone acetylation and deacetylation are also involved in transcriptional regulation in ethylene signaling[23–25].

Rice (*Oryza sativa*) is an important monocotyledonous crop and usually lives in water-saturated soil in most of the life cycle. Despite the essential role ethylene plays in the adaptive responses of rice to hypoxia stress, the ethylene signaling pathway in rice has not been systematically studied. Genes homologous to the *Arabidopsis* ethylene signaling components have been identified and some are characterized in rice, including ethylene receptor gene *OsETR2*, *RTE1*-like gene, *CTR1*-like gene, *EIN2*-like gene and *EIN3*-like gene[26–30]. Adopting an effective screening system, we have isolated a set of rice ethylene-response mutants[31]. The analyses of these mutants suggest that ethylene signaling in rice and *Arabidopsis* has both conserved and divergent aspects[32–35].

Histidine kinases (HK) play crucial roles in the regulation of plant development in response to hormones, as well as environmental stimuli[36,37]. HK-mediated multistep phosphorelay involves hybrid-type HK with both histidine kinase and receiver domains, His-containing phosphotransfer protein (HPt), and response regulator (RR)[37,38]. Ethylene receptors are structurally similar to bacterial HKs and some receptors such as ETR1 and ERS1 do have canonical HK activity[39,40]. However, the HK activity of ethylene receptor is not required for ethylene signaling but only plays a modulating role in the pathway[41,42]. So far, how the ethylene receptor transmits signals remains largely unclear. It has been reported that a non-ethylene receptor HK, *Arabidopsis* authentic HK5 (AHK5), acts as a negative regulator in the ETR1 dependent signaling pathway in which ethylene and ABA inhibit the root elongation[43]. In contrast, the maize homolog ZmHK9 acts as a positive regulator in the root growth response to ethylene and ABA in transgenic *Arabidopsis*[44]. OsHK1, a rice histidine kinase[38], is reported to play roles in root growth and circumnutations through a cytokinin-related pathway[45]. In these studies, however, little is known about the molecular mechanism by which the HKs regulate the signaling cascade.

In this study, we characterized the rice root-specific ethylene-insensitive mutant *mhz1* and found that *MHZ1* encodes the rice histidine kinase OsHK1. MHZ1 positively modulates ethylene response in rice roots. Biochemical analysis showed that MHZ1 is a functional hybrid-type HK, which autophosphorylates in a conserved histidine and transfers the phosphoryl group via its receiver domain to OsAHP1/2 and then further to response regulator OsRR21. Genetic evidence demonstrates that the HK activity of MHZ1 and it-mediated phosphorelay are required for regulation of root ethylene response in rice. More interestingly, we discover that the ethylene receptors, via GAF domain, can directly bind to MHZ1 protein and inhibit its kinase activity based on both in vitro and in vivo analyses. These findings reveal a previously unidentified mechanism for the ethylene receptor signal transduction.

## Results

**Characterization of *mhz1* and gene identification**. We have isolated a set of rice ethylene-response mutants and the *mhz1* exhibited root-specific ethylene-insensitive phenotype[31]. In air, etiolated seedlings of two allelic mutants *mhz1-1* and *mhz1-2* were very similar in coleoptile/shoot and root growth to WT. In ethylene, WT root length was drastically reduced whereas *mhz1-1* and *mhz1-2* root growth was not inhibited, indicating a complete ethylene-insensitive phenotype in primary roots of the two mutants (Fig. 1a). Coleoptile growth of *mhz1-1* and *mhz1-2* responded normally to ethylene, except that the mutants have slightly longer coleoptiles than WT (Fig. 1a). Light-grown *mhz1-1* seedlings had longer roots than WT (Supplementary Fig. 1a). Two additional allelic mutants (*mhz1-3* and *mhz1-4*) were further identified and they resembled *mhz1-1* and *mhz1-2* in ethylene responses (Supplementary Fig. 1b). These results indicate that *mhz1* is insensitive to ethylene in root growth.

The *MHZ1* gene was identified to be LOC_Os06g44410 through TAIL-PCR analysis and the T-DNA was inserted in the fifth intron between 2032 bp and 2033 bp from the start codon of the gene in *mhz1-1* (Fig. 1b, c). No *MHZ1* expression was detected in *mhz1-1* (Fig. 1d). Other alleles were further analyzed (Fig. 1b, c and Supplementary Fig. 1b). Genetic transformation with the WT genomic DNA fragment rescued the ethylene-insensitive phenotype of *mhz1-1* (Supplementary Fig. 1c). All these results indicate that *MHZ1* corresponds to the locus LOC_Os06g44410, which encodes the histidine kinase OsHK1[38]. The *mhz1-1*, *-2*, *-3*, *-4*, *-5* mutants may be renamed as *Oshk1-4*, *-5*, *-6*, *-7*, *-8* following the *Oshk1* mutants identified by Lehner et al.[45]. For simplicity, the original mutant names were used since these mutants have been named as *mao huzi* (*mhz*, Chinese name with an English meaning of cat whiskers)[31].

*MHZ1/OsHK1* encodes a histidine kinase of 936 amino acids, with a histidine kinase domain (amino acids 365 to 655) and a receiver domain (amino acids 817 to 961) (Fig. 1c and Supplementary Fig. 2a). Phylogenetic analysis indicates that MHZ1 does not belong to ethylene receptor family or cytokinin receptor homologs (Supplementary Fig. 3). MHZ1 is homologous to AHK5 and ZmHK9. Both of the genes were reported to play roles in modulating ethylene responses[43,44]. MHZ1 is clustered with homologous proteins from monocotyledonous plants (Supplementary Fig. 2b). Coiled-coil and PAS domains were noted in the N-terminal end of these proteins (Supplementary Fig. 2c).

**MHZ1 overexpression enhances ethylene response in roots**. To study the gene function of *MHZ1*, we overexpressed *MHZ1* in WT rice (Fig. 1e, f and Supplementary Fig. 4d). Compared with the WT, the high-expression *MHZ1-OE* lines had shorter roots both in air and in ethylene, indicating a constitutive ethylene-response phenotype (Fig. 1e). The short root phenotype of *MHZ1-OE* lines was not due to elevation of ethylene emission since the ethylene production was not enhanced in these lines (Supplementary Fig. 5). Treatment with 1-methylcyclopropene (1-MCP)[46] largely inhibited the short root phenotype of these lines, indicating that ethylene receptor signaling is required for MHZ1 function (Fig. 1e).

We further examined expression of ethylene-responsive genes identified in our previous studies[31,32,34,35]. qPCR analysis showed that ethylene-induction of *OsRRA5*, *OsERF002* and *OsRAP2.8* expression was largely blocked in *mhz1*, whereas the expression of these genes was substantially enhanced in roots of

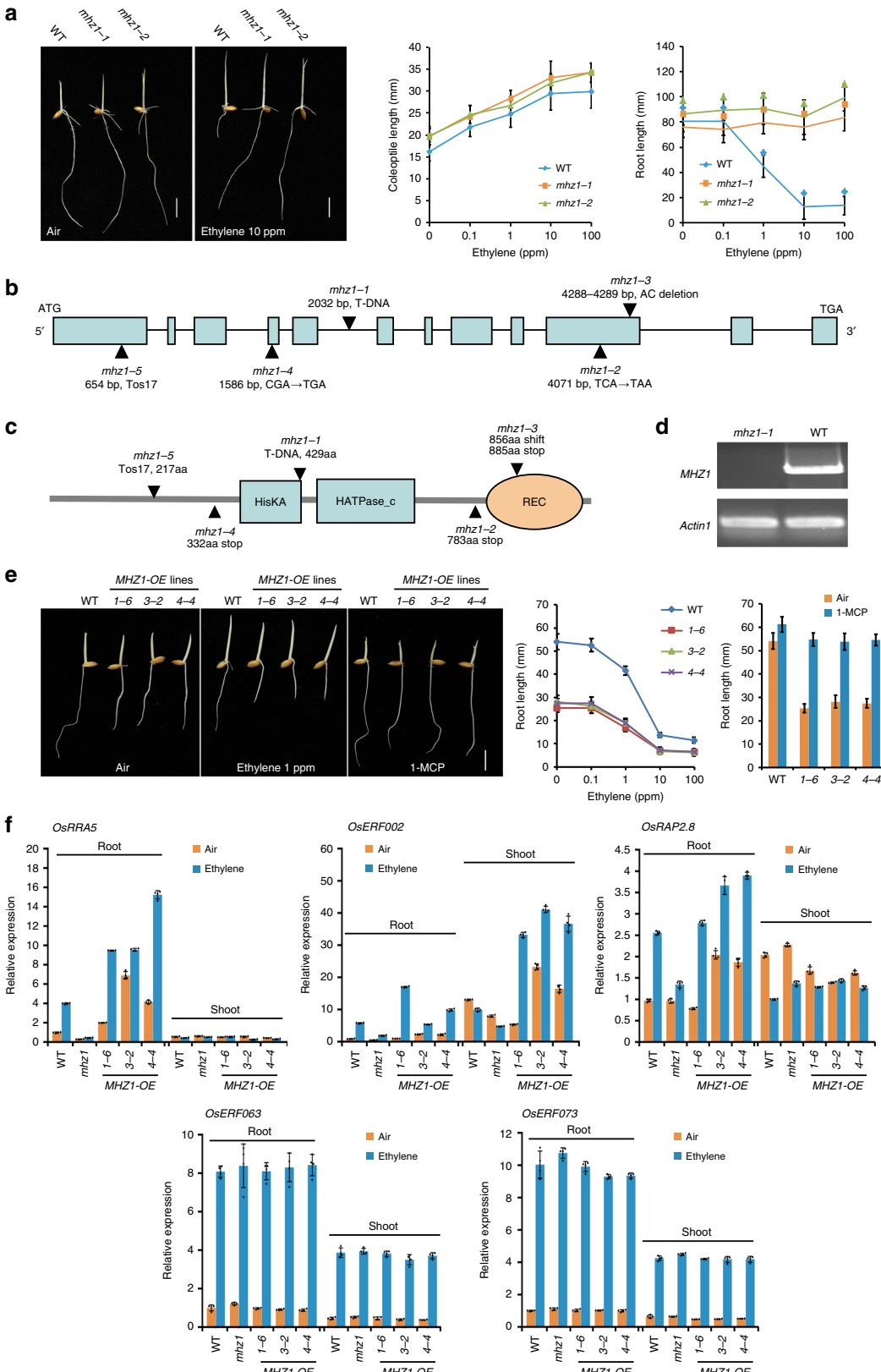

MHZ1-overexpressing plants in the presence or absence of ethylene (Fig. 1f). In shoots of MHZ1-overexpressing plants, OsRRA5 and OsRAP2.8 expression was not enhanced in air or ethylene; whereas OsERF002 expression was promoted compared to the corresponding WT levels especially in the presence of ethylene (Fig. 1f). The ethylene induction of OsERF063 and

OsERF073 was not affected in both roots and shoots of mhz1 or MHZ1-OE plants, suggesting that expression of some genes is independent of MHZ1 (Fig. 1f). These results further confirmed the different ethylene responsiveness of mhz1 mutant and MHZ1-OE plants at molecular levels with some organ and gene specificity.

**Fig. 1 MHZ1 positively regulates the ethylene response in rice roots. a** Ethylene response phenotype of *mhz1* alleles. Etiolated seedlings were treated with various concentrations of ethylene in darkness. Representative seedlings grown in the air and in 10 ppm ethylene are shown (Left). Coleoptile (Center) and root lengths (Right) are means ± SD, *n* > 30. Bars indicate 10 mm. **b** *MHZ1* genomic structure and mutation sites of different *mhz1* alleles. Colored boxes indicate exons and horizontal lines indicate introns. **c** Schematic structure of MHZ1 and mutation sites of different *mhz1* alleles. **d** *MHZ1* gene expression in WT and *mhz1-1. Actin1* was amplified as internal control. **e** *MHZ1* overexpression lines (*MHZ1-OE*) have constitutive ethylene response. *MHZ1* native promoter was used to drive the *MHZ1* cDNA for overexpression. Etiolated seedlings were treated with various concentrations of ethylene and 10 ppm 1-MCP under darkness. Bars indicate 10 mm. Root lengths (Right) are means ± SD, *n* > 30. **f** Expression of ethylene-inducible genes *OsRRA5, OsERF002 OsRAP2.8, OsERF063* and *OsERF073* in *mhz1-1* and *MHZ1-OE* lines compared with WT as revealed by qPCR. Data are means ± SD, *n* = 4. Source data are provided as a Source Data file.

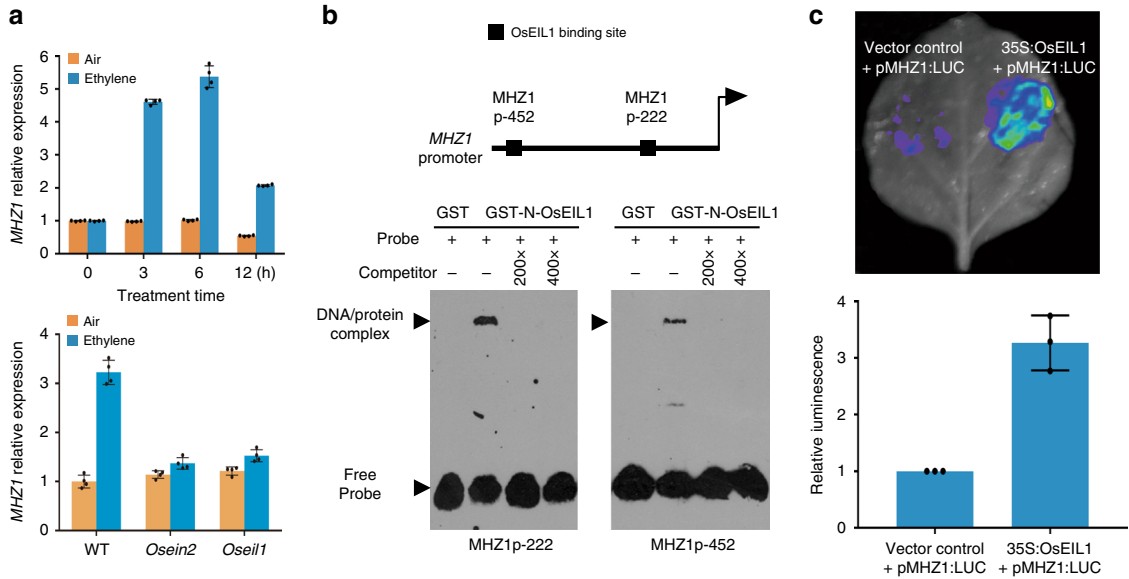

**Fig. 2 *MHZ1* expression is induced by ethylene through directly binding of OsEIL1 to its promoter region. a** *MHZ1* expression induced by ethylene is dependent on OsEIN2 and OsEIL1. Two-day-old etiolated seedlings of WT or *Osein2* and *Oseil1* were treated with 10 ppm ethylene or air. Data are means ± SD, *n* = 4. **b** Binding of OsEIL1 protein to the OsEIL1 binding site (EBS) containing region of *MHZ1* promoter in vitro. GST-tagged OsEIL1 N-terminus protein was isolated and incubated with biotin-labeled probes carrying EBSs of *MHZ1* promoter. An excess of unlabeled probe was added as the competitor. **c** Transient expression of OsEIL1 in tobacco leaves activated *MHZ1* promoter activity. Relative luminescence intensity was calculated, data are means ± SD from three biological replicates. Source data are provided as a Source Data file.

**Ethylene-induced *MHZ1* expression requires OsEIN2 and OsEIL1.** *MHZ1* transcripts are abundant in roots but less in coleoptiles and other organs, and are induced by ethylene in roots (Fig. 2a and Supplementary Fig. 4a). The ethylene induction of *MHZ1* requires OsEIN2 and OsEIL1, and OsEIL1 can bind to the ATGTA elements in the *MHZ1* promoter and activate the promoter activity in a tobacco transient assay system (Fig. 2a, b, c). Promoter-GUS analysis further reveals that *MHZ1* promoter activity is mainly localized in root initiation sites at node, root vascular cylinder and root cortex. The activity is also observed in stem, leaf, grain hull and coleoptile (Supplementary Fig. 4b). Ethylene treatment mildly enhanced the *MHZ1* promoter activity especially in the region immediately above the meristem tissue of root tip (Supplementary Fig. 4c).

**MHZ1 transfers phosphoryl groups to OsRR21 via OsAHPs.** Next, we examined whether MHZ1 has HK activity. Different mutant proteins or truncated versions were produced and tested for atuophosphorylation ability (Fig. 3a). The GST-MHZ1 protein (amino acids 365 to 968) containing the kinase domain and receiver domain displays strong kinase activity in the presence of $Ca^{2+}$ in our phosphorylation assay, and this activity was abolished when the conserved His at 375 position was mutated to Gln (Fig. 3b). A smaller radioactive band was noted below the normal GST-MHZ1, likely representing a degradation product (Fig. 3b).

At the physiological level of ATP concentration, GST-MHZ1 had kinase activity in the presence of $Ca^{2+}$ or $Mg^{2+}$ (Supplementary Fig. 6a). These results indicate that MHZ1 is a functional HK.

When the receiver domain was removed, GST-KD or MBP-KD containing kinase domain (amino acids 365 to 655) still had autophosphorylation activity (Fig. 3c). When the conserved G1 or G2 box for ATP binding in the kinase domain was mutated in MBP-G1(G588A, G590A) or MBP-G2(G618A, G620A), the kinase activity was disrupted (Fig. 3c). However, the two mutated proteins can be phosphorylated by the normal kinase domain GST-KD (Fig. 3c), suggesting that MHZ1 phosphorylation can occur in *trans* manner.

Next, we investigated whether MHZ1 can transfer phosphoryl groups to putative downstream HPt and B-type RR components in rice. Rice has five HPts (OsAHP1/2, OsPHP1/2/3) and more B-type RRs[47,48]. Results showed that GST-MHZ1 can transfer phosphoryl groups to OsAHP1 and OsAHP2 instead of OsPHP1 or OsPHP2 (Fig. 3d), possibly implying substrate specificity. The OsPHP3 was not tested because *OsPHP3* is a more diverged pseudogene[47]. H79Q and H80Q mutations in OsAHP1 and OsAHP2, respectively, disrupted their phosphorylation by GST-MHZ1 (Fig. 3e), suggesting that these residues are likely to be the phosphoryl-accepting sites.

Among the rice B-type response regulators, we selected the originally identified six (OsRR21 to OsRR26)[47] to express in *E. coli*. Only OsRR21 and OsRR26 were successfully expressed, and

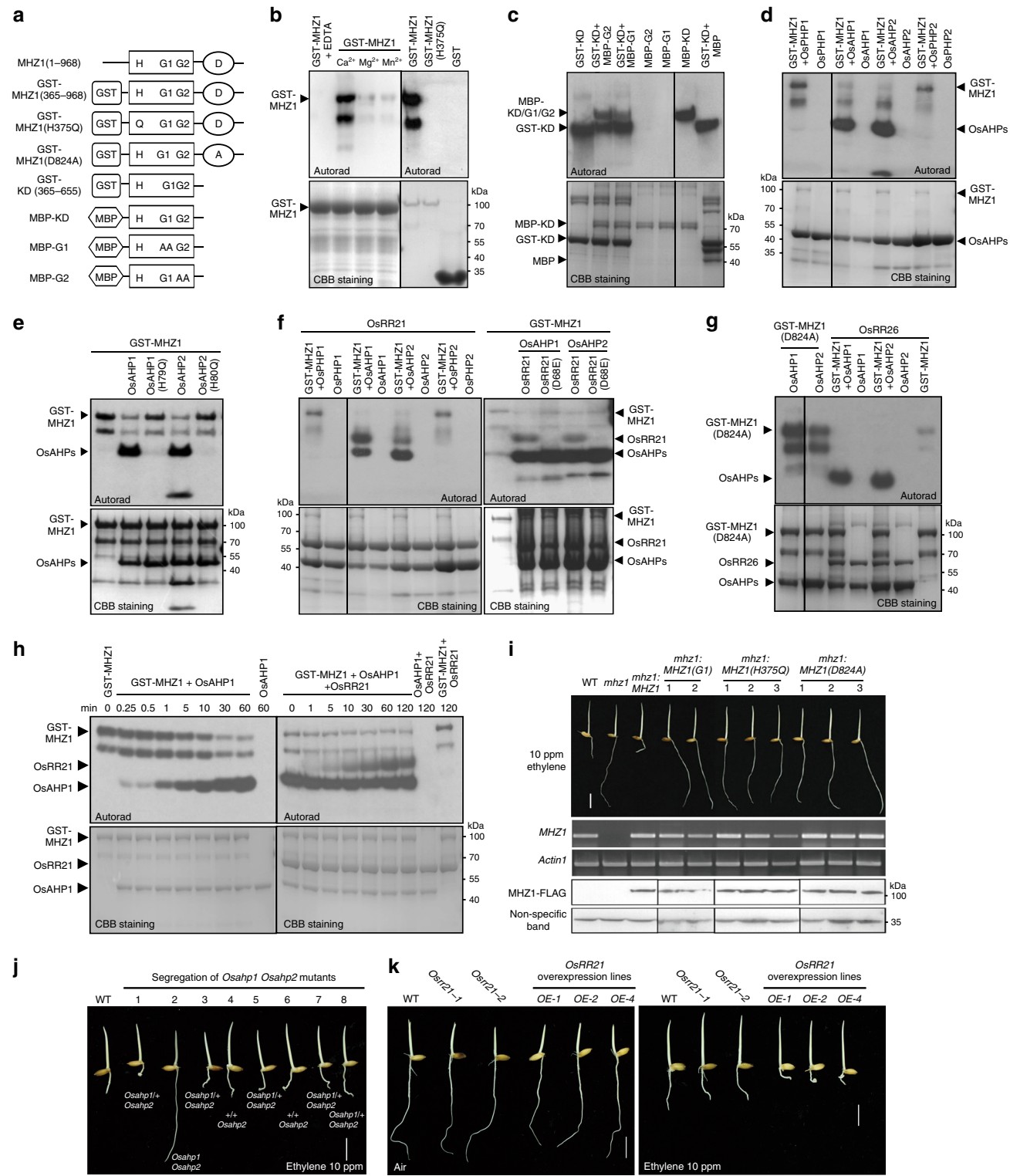

the OsRR21 but not OsRR26 can accept the phosphoryl group transferred from the phosphorylated OsAHP1 or OsAHP2 (Fig. 3f, g), suggesting substrate specificity. D68E mutation of OsRR21 abolished its phosphorylation (Fig. 3f, right panel), suggesting that D68 may be the phosphoryl-accepting site. D824A mutation in the receiver domain of GST-MHZ1 disabled the phosphorelay from MHZ1 to OsAHP1 or OsAHP2 (Fig. 3g, the left two lanes), indicating that the D824 residue of MHZ1 is indispensable for phosphotransfer from GST-MHZ1 to OsAHP1 or OsAHP2. All these results support that MHZ1 can

autophosphorylate and transfer the phosphoryl group via its receiver domain to OsAHP1/OsAHP2 and then further to OsRR21 through phosphorelay.

Time-course analysis of the phosphorelay between GST-MHZ1, OsAHP1/2 and OsRR21 was performed to verify the MHZ1-mediated phosphorelay system. GST-MHZ1 was autophosphorylated first in the presence of [γ-$^{32}$P]ATP before being incubated with OsAHP1. Over the time course, GST-MHZ1 autophosphorylation level was gradually decreased whereas the OsAHP1 phosphorylation was steadily enhanced (Fig. 3h, left

**Fig. 3 MHZ1-mediated phosphorelay pathway is required for ethylene-inhibited root growth. a** MHZ1 and its truncated/mutant versions used for phosphorylation analysis. H, G1, G2, and D indicate conserved residues or boxes. MBP indicates maltose-binding protein. **b** MHZ1 has histidine kinase activity. The radioactive band below the GST-MHZ1 indicates a degradation product. **c** Trans-phosphorylation between MHZ1 molecules. MBP-G1 has G588A and G590A mutations at G1 box. MBP-G2 has G618A and G620A mutations at G2 box. **d** MHZ1 can transfer its phosphoryl groups to OsAHP1 and OsAHP2 rather than OsPHP1 or OsPHP2. **e** Phosphorelay from MHZ1 to OsAHP1 and OsAHP2 was abolished when the conserved histidine was mutated in OsAHP1/2. **f** MHZ1 can transfer its phosphoryl groups to OsAHP1/2 and further to OsRR21. D68E mutation in OsRR21 disrupted its phosphoryl-accepting ability. **g** D824A mutation in MHZ1 receiver domain blocked its phosphorelay to OsAHP1/2, and OsRR26 cannot accept phosphoryl groups transferred from MHZ1 and OsAHP1/2. **h** Time course of the phosphorelay from MHZ1 to OsAHP1 (left panel) and further to OsRR21 (right panel). After the reaction in the left panel was finished, the OsRR21 was added and incubated for various times in the right panel. **i** MHZ1 but not its mutant versions rescued the ethylene-insensitive phenotype of mhz1 (mhz1-1) mutant. cDNAs of MHZ1 and its mutant versions MHZ1(G1), MHZ1(H375Q) and MHZ1(D824A), fusioned with a 3 × FLAG sequence, driven by MHZ1 native promoter, were transformed into the mhz1-1 to observe the root ethylene response. Total proteins of each line were immunoblotted for MHZ1-FLAG with anti-FLAG antibody. A non-specific band was used as a loading control. MHZ1 gene expression was examined by RT-PCR and Actin1 was amplified as control. Bar indicates 10 mm. **j** Ethylene response of Osahp1 Osahp2 double-mutant. Osahp1 Osahp2 double-mutant was segregated from the self-bred progenies of an Osahp1 (heterozygous)/Osahp2 (homozygous) plant. " + " indicates wild-type OsAHP1. **k** Ethylene response of OsRR21 mutants and overexpression lines (OE). For **j** and **k**, etiolated seedlings were treated with 10 ppm ethylene or air for 2.5 days. Bars indicate 10 mm. Source data are provided as a Source Data file.

panel), suggesting an active phosphotransfer from GST-MHZ1 to OsAHP1. After phosphotransfer from GST-MHZ1 to OsAHP1, the OsRR21 was further added to the assay system and its phosphorylation level was also increased (Fig. 3h, right panel). By contrast, as negative controls, GST-MHZ1 plus OsRR21, or OsAHP1 plus OsRR21 did not result in OsRR21 phosphorylation (Fig. 3h, right panel). Similar phosphotransfer also happened from GST-MHZ1 to OsAHP2 and further to OsRR21 (Supplementary Fig. 6b). Altogether, all these data indicate that the activated GST-MHZ1 could transfer its phosphoryl group to OsAHP1/OsAHP2 and further to the downstream RRs, e.g., OsRR21.

**Ethylene signaling requires MHZ1-mediated phosphorelay.** We further analyzed whether MHZ1 kinase activity and it-mediated phosphorelay is essential for ethylene response in rice roots. The coding region of MHZ1 gene tagged with FLAG and driven by MHZ1 native promoter, was mutated in the G1 box (G588A, G590A), the H375 (H375Q) or the D824 (D824A) sites and transformed into mhz1-1. No ethylene response was observed in the roots of homozygous transgenic lines harboring the mutated MHZ1 genes, although the MHZ1-FLAG protein was detected in each homozygous line (Fig. 3i). As a positive control, the roots of homozygous ProMHZ1:MHZ1-FLAG transgenic line in mhz1-1 background displayed normal ethylene response (Fig. 3i). These results indicate that MHZ1 kinase activity, the conserved H375 phosphorylation site and D824 phosphoryl-accepting site are all required for its function in the regulation of root ethylene response in rice.

Since OsAHP1/2 and OsRR21 accept phosphoryl groups from MHZ1, we tested whether they are involved in ethylene signaling. Mutants of OsAHP1, OsAHP2 and OsRR21 were generated through CRISPR/Cas9 and their ethylene responses were examined (Fig. 3j, k and Supplementary Fig. 7a, b, c, d). Results showed that while Osahp1 and Osahp2 single mutants had normal ethylene responses (Supplementary Fig. 7a), Osahp1 Osahp2 double-mutant, which was segregated from the self-bred progenies of the Osahp1 (heterozygous)/Osahp2 (homozygous) plant, exhibited ethylene-insensitive root growth (Fig. 3j and Supplementary Fig. 7c), suggesting that OsAHP1 and OsAHP2 may play redundant roles in ethylene signal transduction. Transgenic plants overexpressing OsRR21 exhibited shorter roots compared with WT both in air and in ethylene (Fig. 3k and Supplementary Fig. 7b), and expression of ethylene-responsive genes are enhanced in these overexpression lines compared with WT (Supplementary Fig. 7e), suggesting that OsRR21 positively modulates ethylene response. Meanwhile,

two mutant lines of OsRR21 exhibited normal ethylene responses (Fig. 3k and Supplementary Fig. 7b, d), indicating that response regulators may play redundant roles in regulating ethylene response. In WT protoplast, OsRR21 enhanced OsRAP2.8 promoter activity and this enhancement is abolished in the protoplast of mhz1, suggesting that OsRR21 function requires signal from MHZ1 (Supplementary Fig. 7f). Ethylene induction of several other response regulator genes suggests that these additional genes may also contribute to regulation of ethylene response (Supplementary Fig. 7g). Altogether, these results indicate that the MHZ1-AHP1/2-OsRR21 phosphorelay pathway is required for ethylene response in rice roots.

**MHZ1 genetically acts at OsERS2.** Genetic analyses were performed to study how MHZ1 interacts with the canonical ethylene signaling pathway. Double-mutant analysis showed that the ethylene hypersensitivity in the roots of Osers2 and Osetr2 ethylene receptor loss-of-function mutants was abolished by mhz1 mutation, suggesting that MHZ1 is required for the ethylene-response phenotype of the receptor mutants.

Next, we used a dominant OsERS2 gain-of-function mutant $Osers2^d$ to test its genetic interaction with MHZ1. $Osers2^d$ was identified as mhz12 from our ethyl methanesulphonate (EMS)-mutagenized population and harbors a dominant mutation (A32V) at the transmembrane domain of OsERS2, which is equivalent to Arabidopsis etr1-3[49] (Supplementary Fig. 8). The mutant showed ethylene-insensitive phenotype (Fig. 4b and Supplementary Fig. 8). The $Osers2^d$ mutation was introduced into MHZ1 overexpression background by crossing and by transgenic approach, and this introduction resulted in complete masking of the constitutive and ethylene-induced short root phenotype of MHZ1-OE plants, without altering the abundance of MHZ1 protein (Fig. 4b, c). These results indicate that gain-of-function mutation of $Osers2^d$ suppressed MHZ1 function in root ethylene response. Combining with the fact that mhz1 suppressed the short root phenotype of Osers2, we propose that MHZ1 may genetically function at the OsERS2 level or they may form a complex.

**OsERS2 interacts with MHZ1 to inhibit its kinase activity.** Genetically, we demonstrated that MHZ1 and OsERS2 function at the same level. Given that they are all HKs in structure (Fig. 4b, c, d), we tested the possibility of protein–protein interaction between MHZ1 and OsERS2. Membrane-based yeast two-hybrid assay showed that yeast cells co-expressing OsERS2-Cub and NubG-MHZ1 were able to grow well on the selective media,

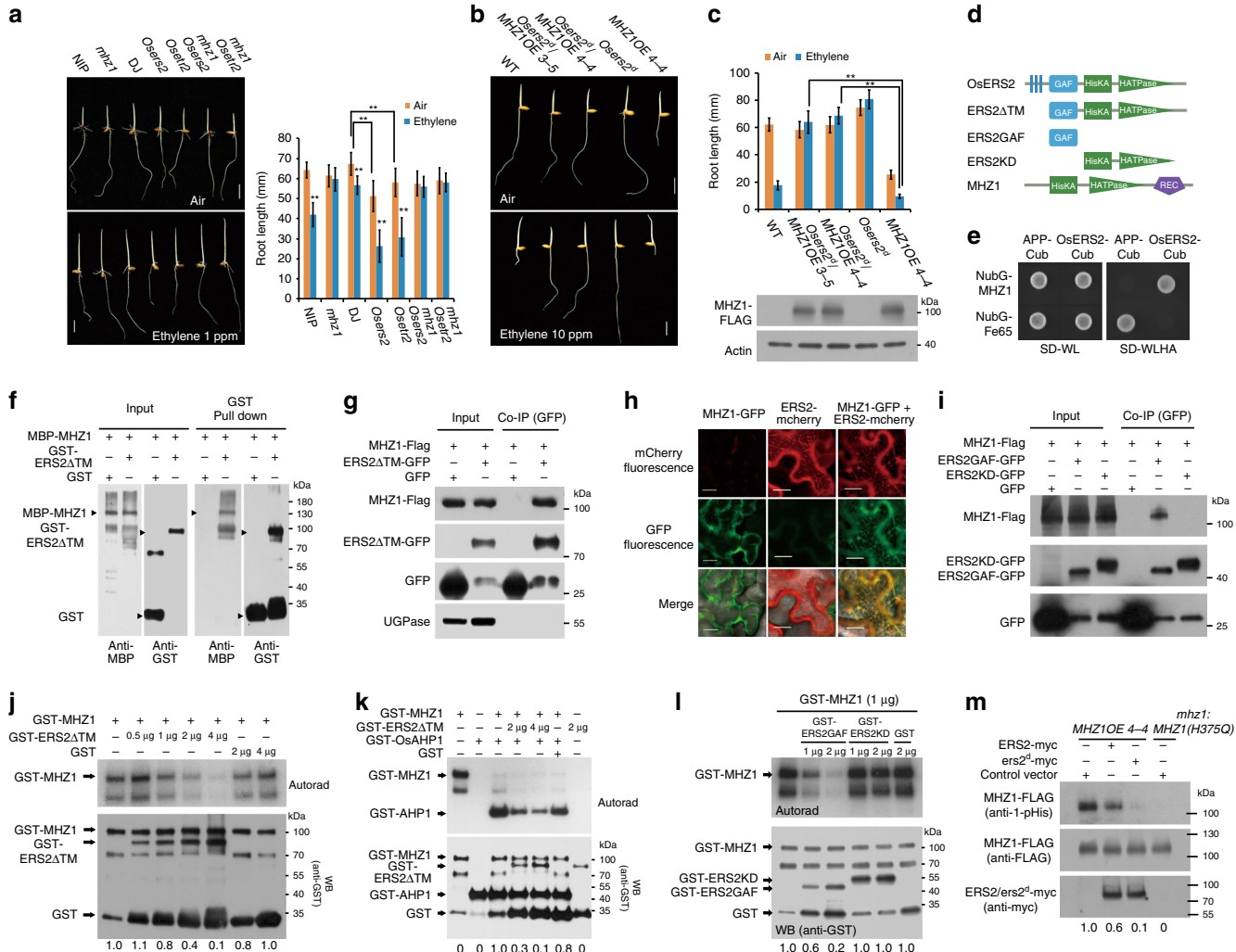

**Fig. 4 MHZ1 physically interacts with and inhibited by ethylene receptors. a** *mhz1* suppressed the ethylene hypersensitivity of *Osetr2* (Dongjin) and *Osers2* (Dongjin) loss-of-function mutants. (Left) Seedlings were treated with air or 1 ppm ethylene. (Right) Data are means ± SD, *n* > 30. \*\**P* < 0.01; Student's *t*-test. DJ, Dongjin; Nip, Nipponbare. Bars indicate 10 mm. **b** Ethylene response of *Osers2d/MHZ1-OE* lines. Etiolated seedlings were treated with air or 10 ppm ethylene. Bars indicate 10 mm. **c** Root length quantification of each line in **b**. Total proteins of each line were immunoblotted for MHZ1-FLAG with anti-FLAG antibody. Data are means ± SD, *n* > 30. \*\**P* < 0.01; Student's *t*-test. **d** Schematic structures of MHZ1 and OsERS2 and their truncated versions. **e** Split-ubiquitin Y2H assay for interaction of MHZ1 and OsERS2. Combination of pTSU2-APP and pNubG-Fe65 (provided in the kit) was used as a positive control. **f** Pull-down of MBP-MHZ1 with GST-OsERS2ΔTM in *E. coli*. **g** Co-IP assays for interaction of MHZ1 and OsERS2. The constructs were cotransformed into rice protoplasts. Total proteins were immunoprecipitated with GFP-Trap and immunoblotted with anti-GFP, anti-FLAG, and anti-UGPase antibodies. **h** OsERS2 facilitates the ER membrane localization of MHZ1. MHZ1-GFP and OsERS2-mCherry proteins were transiently expressed in tobacco leaf epidermal cells. Bars indicate 20 μm. **i** Co-IP assays for interaction domain mapping of MHZ1 and OsERS2. The constructs were cotransformed into rice protoplasts. Total proteins were immunoprecipitated with GFP-Trap and immunoblotted with anti-FLAG, anti-GFP antibodies. **j** OsERS2 inhibits MHZ1 kinase activity. GST was added as a control. Values at the bottom indicate relative phosphorylation levels of GST-MHZ1. **k** OsERS2 inhibited phosphate groups relayed to OsAHP1. Values at the bottom indicate relative phosphorylation levels of GST-AHP1. **l** GAF domain of OsERS2 inhibits MHZ1 kinase activity. GST was added as a control. Values at the bottom indicate relative phosphorylation levels of GST-MHZ1. **m** Histidine phosphorylation of MHZ1 is suppressed by OsERS2 and Osers2d. Vectors carrying *OsERS2-myc* and *Osers2d-myc* were transformed into protoplasts of *MHZ1OE 4-4*. Total protein was immunoprecipitated with anti-FLAG affinity gel and immunoblotted with anti-FLAG or anti-1-p-His (Millipore, MABS1330) antibodies. Values at the bottom indicate relative phosphorylation levels of MHZ1. Protoplasts of ProMHZ1:MHZ1(H375Q) transgenic line transformed with the control vector were used as negative control. Source data are provided as a Source Data file.

suggesting that MHZ1 interacts with OsERS2 in yeast cells (Fig. 4e). GST pull-down assay and coimmunoprecipitation (Co-IP) assay in rice protoplasts further reveals that OsERS2 can interact with MHZ1 both in vitro and in rice cells (Fig. 4f, g). Since MHZ1 lacked the transmembrane domain and is predicted to be a cytoplasmic protein, we examined whether its interaction with OsERS2 could help it localize to the ER membrane through the membrane recruitment assay (MeRA)[50]. MHZ1-GFP and OsERS2-mCherry proteins were transiently expressed in tobacco

leaf epidermal cells and fluorescence was examined. When expressed separately, OsERS2-mCherry was mainly detected in a reticular network-like structure, while MHZ1-GFP was mainly detected in the cytoplasm. When expressed together, MHZ1-GFP was found to co-localize with OsERS2-mCherry, suggesting that OsERS2 facilitated the ER membrane localization of MHZ1 (Fig. 4h). In addition, protein fractionation assay shows that quite amounts of MHZ1 protein were detected in the membrane fractions especially in the presence of gain-of-function Osers2d,

further supporting the association of MHZ1 with membrane-bound OsERS2 (Supplementary Fig. 9a). Consistently, 1-MCP treatment caused abundance of MHZ1 in membrane fraction, whereas ethylene treatment led to decrease of MHZ1 in membrane fraction (Supplementary Fig. 9b). Furthermore, Osers2[d] appeared to have stronger interaction with MHZ1 than wild-type OsERS2 in yeast two-hybrid assay (Supplementary Fig. 9c). To examine which domain of OsERS2 mediates the interaction with MHZ1, we generated truncated versions of OsERS2 (Fig. 4d). Co-IP assay shows that the GAF domain, but not the kinase domain (KD) of OsERS2, is actually responsible for the interaction of OsERS2 with MHZ1 (Fig. 4i).

Next, we examined the effects of this interaction on MHZ1 activity. In the phosphorylation assay, addition of increasing amount of OsERS2 (GST-ERS2ΔTM) drastically reduced the MHZ1 histidine kinase activity, whereas inclusion of GST itself did not significantly affect this activity (Fig. 4j), indicating that OsERS2 can inhibit the autophosphorylation of MHZ1 in vitro. Since autophosphorylated MHZ1 can transfer its phosphoryl group to OsAHPs (Fig. 3), we investigated whether OsERS2 could affect this process. Compared with GST, addition of the OsERS2 apparently reduced the OsAHP1 phosphorylation by MHZ1-mediated phosphorelay (Fig. 4k). The inhibitory effect of OsERS2 on MHZ1 kinase activity and the phosphorelay may not be due to an competitive binding of [γ-$^{32}$P]ATP, as suggested by our results that OsERS2 only had very limited kinase activity in the presence of $Ca^{2+}$ (Fig. 4k, right-most panel, Supplementary Fig. 10a). Given that the GAF domain mediates the interaction of OsERS2 with MHZ1, we examined the effect of GAF domain on MHZ1 autophosphorylation. Phosphorylation assay showed that the GAF domain exerted an inhibitory effect on MHZ1 autophosphorylation while the kinase domain of OsERS2 (GST-ERS2KD) did not show a significant effect (Fig. 4l). All these results indicate that OsERS2 can inhibit both MHZ1 autophosphorylation and MHZ1-mediated phosphorelay likely via its GAF domain.

To examine whether OsERS2 inhibits MHZ1 phosphorylation in vivo, we transfected vectors harboring the OsERS2-myc and Osers2[d]-myc into rice protoplasts isolated from MHZ1OE 4-4 line. With similar MHZ1-FLAG protein levels in each material group (Fig. 4m, middle panel), the histidine phosphorylation level of the MHZ1-FLAG protein is significantly and differentially reduced by expressing OsERS2 or Osers2[d] compared to vector control as revealed by immunoblot analysis using the anti-1-p-His antibody (Fig. 4m, top panel). While OsERS2 and Osers2[d] had similar protein levels (Fig. 4m, bottom panel), Osers2[d] had an apparently stronger inhibitory effect on MHZ1 phosphorylation than wild-type OsERS2 does, probably due to a stronger interaction of Osers2[d] with MHZ1 compared with wild-type OsERS2 (Supplementary Fig. 9a, b, c). In mhz1:MHZ1(H375Q) plant cells, no signal of MHZ1 histidine phosphorylation was detected. Transfection of MHZ1-FLAG into WT and Osers2[d] protoplasts further revealed that Osers2[d] had stronger inhibitory effect on MHZ1 histidine phosphorylation than the control (Supplementary Fig. 9d). All these results clearly demonstrate that OsERS2 and Osers2[d] inhibit MHZ1 histidine phosphorylation in rice cells.

In addition to OsERS2, other ethylene receptors such as OsERS1 and OsETR2 also displayed mild interaction with MHZ1 and moderate inhibitory effect on MHZ1 autophosphorylation (Supplementary Fig. 10b, c, d), suggesting similar roles of ethylene receptors in modulating MHZ1 kinase activity.

**Genetic interaction of *MHZ1* with *OsEIN2*.** We further examined the genetic relation of *MHZ1* with *OsEIN2*. Rice *Osein2* mutant is insensitive to ethylene in both root and coleoptile growth, and overexpression of OsEIN2 in WT seedlings resulted in strong constitutive and enhanced ethylene responses[31]. We generate the *mhz1/OsEIN2-OE* plants by crossing. *OsEIN2-OE* partially suppressed the ethylene-insensitive root growth of *mhz1* in *mhz1/OsEIN2-OE* seedlings in ethylene (Fig. 5a, b), while the *mhz1/OsEIN2-OE* seedlings still have longer adventitious roots at the node above the mesocotyl compared to the *OsEIN2-OE* seedlings, indicating that activated OsEIN2-mediated signaling pathway can partially restore the ethylene response of *mhz1* mutant (Fig. 5c).

We further crossed *MHZ1-OE* with *Osein2* to generate *Osein2/MHZ1-OE* plants (Fig. 5d). In ethylene, the roots of *Osein2/MHZ1-OE* seedlings are longer than that of *MHZ1-OE* seedlings, but are still significantly shorter compared with its air control (Fig. 5d, e). The results indicate that *Osein2* mutation cannot completely block the enhanced ethylene response conferred by *MHZ1* overexpression, implying that MHZ1 may have the ability to accept signal from upstream components, e.g., ethylene receptors, independent of OsEIN2 function.

To further elucidate the relationship between MHZ1- and OsEIN2-mediated pathways, we compared the downstream ethylene-response genes (ERGs) regulated by MHZ1, OsEIN2, or OsEIL1 in rice roots through transcriptome analysis. RNA-seq data suggested that 85% (719) of MHZ1-dependent ERGs were also regulated by OsEIN2 and OsEIL1 (Fig. 5f and Supplementary Data 1). This is consistent with the fact that *MHZ1* is transcriptionally downstream of OsEIN2 and OsEIL1 (Fig. 2). As MHZ1-dependent ERGs only account for half of OsEIN2-dependent ERGs (Fig. 5f and Supplementary Data 1), we further compared the ethylene responsiveness of *mhz1* and *Osein2* by checking the expression of several ERGs, including *OsRRA5*, *OsRAP2.8*, *OsERF002*, *OsERF063*, and *OsERF073*. Whereas the ethylene induction of all five genes were abolished or hampered in *Osein2*, only *OsRRA5*, *OsRAP2.8*, and *OsERF002* expression was affected by *mhz1* (Supplementary Fig. 11a, b), suggesting that the ethylene responsiveness of *mhz1* and *Osein2* is differential in terms of gene expression. These results suggest that MHZ1-mediated pathway shares a subset of ERGs with the OsEIN2 signaling pathway for regulating root growth in rice, although genetically MHZ1 is partially independent of OsEIN2. A GO analysis for the subset of MHZ1-dependent ERGs in comparison to total ERGs was performed using BiNGO[51]. Results showed that compared with total ERGs, MHZ1-dependent ERGs are mainly enriched in auxin signaling pathway and responses to different stimuli (Supplementary Fig. 11c). This is in line with former findings that ethylene functions upstream of auxin signaling to regulate root growth[52–54], indicating that MHZ1 may be involved in the crosstalk between ethylene, auxin and different stimuli to regulate root growth.

During an effort to screen for the suppressor of *OsEIN2-OE* plants, three suppressors lines (SOE-7407, -410, -9744) were found to be very similar to the *mhz1/OsEIN2-OE* seedlings after ethylene treatment (Supplementary Fig. 12). *MHZ1* was identified to harbor the mutation sites in these suppressors by sequencing (Supplementary Fig. 12). These findings further support the genetic relationship between *MHZ1* and *OsEIN2*.

**Discussion**

Through mutant analysis, we identified MHZ1 as a positive modulator of root ethylene response in rice. MHZ1 has autophosphorylation ability and can transfer its phosphoryl group to OsRR21 via OsAHP1 and OsAHP2. Ethylene receptor OsERS2 physically interacts with MHZ1 to inhibit MHZ1 autophosphorylation and signaling. We propose that in the absence of ethylene, the ethylene receptors are in active conformations, which facilitates their interaction with MHZ1 and MHZ1 kinase

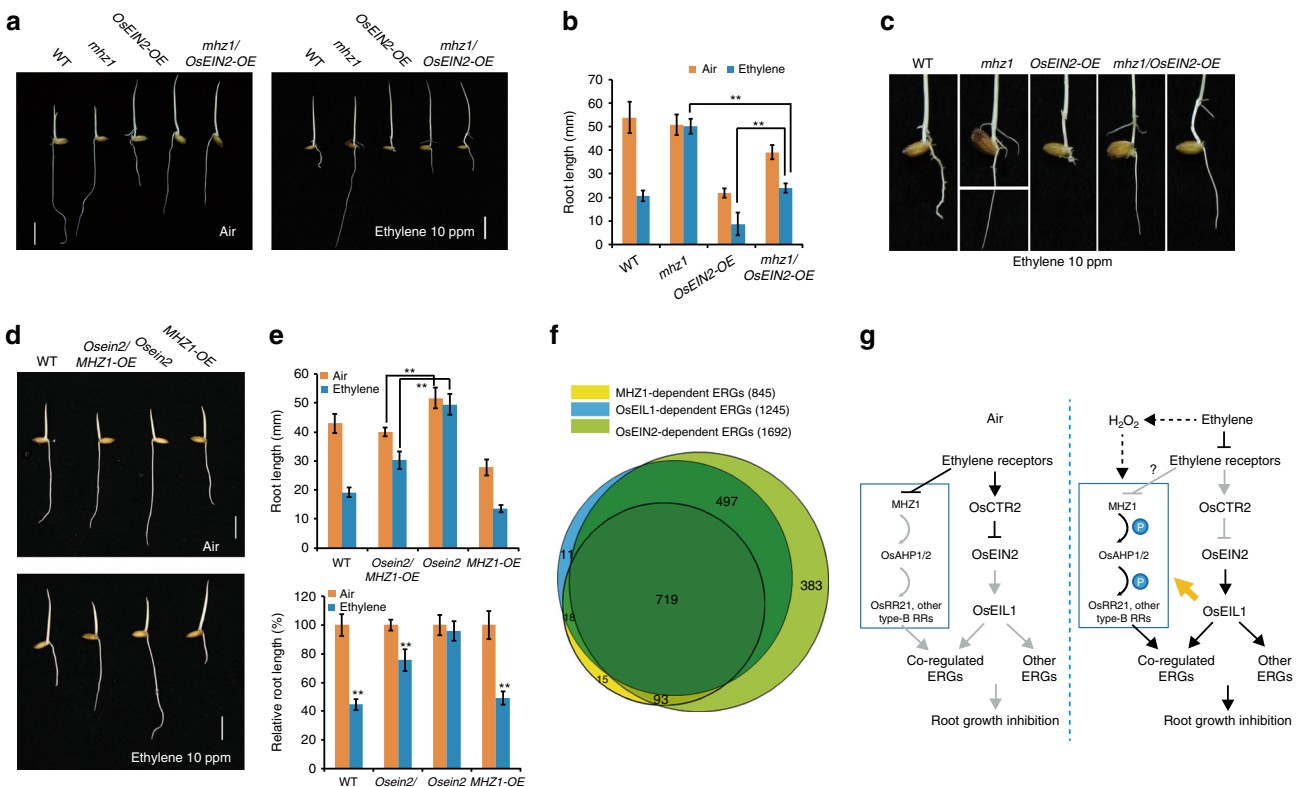

**Fig. 5 Genetic interaction of MHZ1 and OsEIN2-mediated pathways. a** Ethylene response of *mhz1/OsEIN2-OE* in comparison with *mhz1* and *OsEIN2-OE*. Bars indicate 10 mm. **b** Quantification of root length of the mutants in **a**. Root lengths are means ± SD, *n* > 30. **P < 0.01; Student's *t*-test. **c** Enlargement of root features of the mutants treated with 10 ppm ethylene in **a**. **d** Ethylene response of *Osein2/MHZ1-OE* in comparison with *Osein2* and *MHZ1-OE*. Rice seedlings were grown in dark for 2 days with air or 10 ppm ethylene. Bars indicate 10 mm. **e** Quantification of root length (above) and relative root length (below) of the mutants in **d**. Data are means ± SD, *n* > 30. **P < 0.01; Student's *t*-test. **f** Comparison of MHZ1-, OsEIN2-, and OsEIL1-regulated ethylene-response genes. Etiolated seedlings of WT, *mhz1*, *Osein2* and *Oseil1* were grown for 2 days before treated with air or 10 ppm ethylene for 3 h. Roots were subjected to RNA-seq analysis with three biological replicates. Ethylene-response genes (ERGs) were identified in WT according to the gene expression levels with at least relative twofold changes (*q*-value < 0.05) in ethylene treatment compared to those in the air. A total of 1789 ERGs were identified. **g** A proposed working model for MHZ1-mediated ethylene signaling in rice. Rice has the conserved components of ethylene signaling as those in Arabidopsis, including ethylene receptors, OsCTR2, OsEIN2, and OsEIL1. Besides the conserved receptor-CTR2-OsEIN2-OsEIL1 signaling pathway, our results suggests a novel MHZ1-AHP1/2-OsRR21 phosphorelay pathway, through which ethylene receptors could regulate the root growth of rice by suppressing the kinase activity of MHZ1. In the absence of ethylene, ethylene receptors are in active conformations, which may facilitate their interaction with MHZ1 and suppress MHZ1 activity. With ethylene, ethylene receptors possibly released the repression effect on MHZ1 and the phosphorelay pathway is activated. The MHZ1-mediated phosphorelay pathway and the OsEIN2-mediated pathway may work together to regulate a subset of downstream gene to modulate root growth. *MHZ1* and *OsRRs* can also be transcriptionally induced by ethylene through the OsEIN2-mediated pathway, which may facilitate the maintenance of MHZ1-mediated signaling after activation. The orange arrow indicates transcriptional activation. In addition, $H_2O_2$ may also function through MHZ1 to participate in the ethylene-regulated root growth. Source data are provided as a Source Data file.

activity is suppressed. Upon ethylene perception, the receptors may release their inhibition on MHZ1, triggering MHZ1-mediated phosphorelay for regulation of root growth. This pathway may be a branch in parallel with the OsEIN2-mediated one, and both act downstream of ethylene receptor signaling to regulate root growth in rice (Fig. 5g).

Our study reveals a previously unidentified mechanism, by which ethylene receptor OsERS2 binds to the histidine kinase MHZ1 and suppressed its autophosphorylation and phosphorelay, adding knowledge toward how the ethylene receptors transmit signals. This conclusion may be in line with those from bacteria studies. In *Pseudomonas aeruginosa*, virulence is partially controlled by the response regulator protein GacA, which receives phosphoryl group from upstream histidine kinase GacS. The kinase activity of GacS is proved to be inhibited by another histidine kinase RetS through direct binding, and the binding did not require the conserved phosphorylation site in RetS[55–57]. The interaction of the two histidine kinases facilitates the bacteria to

integrate environmental signals to control complex adaptive processes. Plant ethylene receptors are structurally related histidine kinase-like proteins. Although OsERS2 (Supplementary Fig. 10a) and OsETR2 showed kinase activity[29], it is not clear if these kinase activities would have any roles in inhibition of MHZ1 signaling. Actually, the GAF domain in OsERS2 is sufficient for mediating the interaction with MHZ1 and plays a major role in inhibition of MHZ1 kinase activity (Fig. 4). GAF domain has been found in all ethylene receptors and the function has not been identified so far. Our study discloses its function in connecting downstream histidine kinase MHZ1 during ethylene signaling in rice roots. Previously, the N-terminal part (residues 1–349) of ETR1 containing the GAF domain has been found to largely rescue the *ctr1-1* and *ctr1-2* mutant phenotypes, possibly suggesting an alternative pathway bypassing CTR1 in Arabidopsis[58]. A recent research also suggested that ETR1 is involved in the multistep phosphorelay pathway by interacting with AHP proteins through its RD domain[59]. Whether the *Arabidopsis*

AHK5[43], a homolog of MHZ1, is involved in the above pathway needs to be further investigated. Ethylene transcriptionally induces the expression of ethylene receptor genes *ERS1*, *ERS2*, and *ETR2* in *Arabidopsis*[11] and *OsETR2* in rice (Supplementary Data 1). It is possible that ethylene-induced receptor gene expression may function as a desensitizing approach at later stage for ethylene receptor to re-lock MHZ1 after the initial biochemical triggering of the signaling and completeness of ethylene response.

Downstream of ethylene receptors, two branch pathways are proposed (Fig. 5g). Apparently, the conserved OsCTRs-OsEIN2-OsEIL1 pathway should play major roles in both roots and aerial parts, whereas the MHZ1-OsAHP1/2-OsRR21 pathway may specifically play roles in rice roots. Ethylene-induced OsEIN2 accumulation is not affected in *mhz1* or *mhz1/OsEIN2-OE* plants, further supporting a separate role of the MHZ1 in root ethylene response (Supplementary Fig. 13). Transcriptionally, *MHZ1* and *OsRR21* can be induced by the conserved OsEIN2-OsEIL1 pathway and this feature may facilitate maintenance of MHZ1-mediated signaling after activation (Fig. 2 and Supplementary Fig. 7g). In our RNA-seq analysis, 719 ERGs were identified to be shared by MHZ1, OsEIN2 and OsEIL1 (Fig. 5f and Supplementary Data 1). It is proposed that the two pathways may work together to modulate the expression of these ERGs, thus regulating root growth. Interruption of either pathway would abolish the ethylene induction of these ERGs, causing an ethylene-insensitive phenotype of rice roots. On the other hand, when either pathway is interrupted, constitutive activation of the other pathway could partially restore the induction of downstream ERGs.

Two-component systems also participate in cytokinin signaling in *Arabidopsis* and similar mechanism may exist in rice[38,47,60–63]. Considering that rice may have only two functional HPts (OsAHP1 and OsAHP2) for signaling[47,48], it is hence possible that these two genes play roles in both ethylene signaling and cytokinin signaling in specific manners depending on distinct treatments, times, cell types, tissues and/or organs. Actually, only limited number of homozygous *Osahp1 Osahp2* seedlings were identified from 168 self-bred progenies of the *Osahp1* (heterozygous)/*Osahp2* (homozygous) plant, implying that the *Osahp1 Osahp2* double-mutant may have some degree of defect in embryo development, which may be caused by a defect in cytokinin signaling. Other possibilities cannot be excluded. Similar sharing of the histidine-containing phosphotransmitter has been reported in filamentous fungi *Aspergillus nidulans*[64].

*Arabidopsis* has a MHZ1 homolog AHK5. Mutation of the gene caused mildly enhanced ethylene response in *Arabidopsis ahk5* roots[43]. This response is in contrast with the complete ethylene-insensitive response in the present rice *mhz1* root, suggesting that rice may have evolved to adopt the MHZ1 pathway in a different mechanism to strongly control ethylene-regulated root growth for adaptation in water environment. *ahk5* is also sensitive to ABA, and our present *mhz1* is slightly insensitive to ABA (Supplementary Fig. 14), suggesting that MHZ1 may also participate in ABA- or other stress-related processes in rice. A maize homolog *ZmHK9* has been characterized and the gene is highly expressed in roots and induced by drought and ABA treatment[44]. Overexpression of the gene in transgenic *Arabidopsis* plants caused hypersensitivity to ABA and ethephone, and led to drought tolerance through regulation of stomatal density and stomatal closure[44]. These studies suggest that MHZ1 may also have other functions in addition to its roles in ethylene response.

In the N-terminal end of MHZ1, a PAS domain is noted (Supplementary Fig. 2c). This domain is usually involved in ligand binding and/or protein interaction, suggesting other sensing possibilities in addition to be regulated by ethylene receptors. The possibility that MHZ1 serves as a cytoplasmic co-receptor of ethylene signaling cannot be excluded since both MHZ1 and ethylene receptors are histidine kinases or structurally similar proteins. It should be noted that, *Arabidopsis* AHK5, several AHPs and response regulators can form signaling network to modulate stomatal closure in response to $H_2O_2$ and/or ethylene[65–67]. Root growth of *mhz1* is slightly insensitive to $H_2O_2$ (Supplementary Fig. 15), suggesting that $H_2O_2$ may partially function through MHZ1 to inhibit root growth of rice. Given that ethylene induces $H_2O_2$ production in *Arabidopsis*[68], it is possible that $H_2O_2$ may also be involved in the proposed pathway (Fig. 5g). This result is consistent with the GO analysis that MHZ1-dependent ERGs are enriched in auxin signaling pathway and also responses to different stimuli. OsHK1/MHZ1 is previously reported to play roles in rice large radius root tip circumnutations through a cytokinin-related pathway[45]. As ethylene is also demonstrated to stimulate nutations in *Arabidopsis* in an auxin transport-dependent manner[69], it is possible that MHZ1 is also involved in the crosstalk between ethylene and auxin, cytokinin or $H_2O_2$ to regulate root growth.

Collectively, we identified the histidine kinase MHZ1 as a regulator of ethylene signaling in rice. Ethylene receptor OsERS2 interacts with MHZ1 through GAF domain and inhibits MHZ1-mediated signaling. The MHZ1-mediated pathway may function as a branch in parallel with the conserved pathway to regulate root growth especially under semi-aquatic environment (Fig. 5g). Our data provide valuable insights into the mechanism of ethylene signaling in rice and should facilitate improvement of stress adaptation and relevant agronomic traits in crops.

## Methods

**Materials, ethylene treatment, and gene identification.** The rice (*Oryza sativa* L.) mutants *mhz1*, *Osein2/mhz7-1*, *Oseil1/mhz6*, and *Osers2*[d]/*mhz12* mutants were previously identified by Ma[31]. *OsEIN2-OE* lines were generated by Ma[31]. All the *mhz1* alleles are in Nip background, and the *Osers2* and *Osetr2* mutants are in DJ background. Material propagation and crosses were carried out in the Experimental Station of the Institute of Genetics and Developmental Biology in Beijing from May to October of each year. For ethylene-response assay, seeds were soaked at 37 °C for 2 days and the germinated seeds were placed on stainless net for ethylene treatment at 28 °C in dark for 3 days if not specified, with a water level below the seeds[31]. Lengths of roots and/or coleoptiles were measured. For ABA treatment, stock solution of ABA was prepared in ethanol and diluted into solutions of different concentrations with water. Equivalent volumes of ethanol were added to the control. The *MHZ1* gene was identified by TAIL-PCR method. To generate *MHZ1-OE* lines, *MHZ1* CDS driven by the native promoter (3 kb sequence upstream of ATG) was transformed into WT rice and homozygous transgenic lines with higher *MHZ1* expression levels were analyzed. The ethylene receptor loss-of-function mutants *Osers2* and *Osetr2* were purchased and identified through PCR with primers suggested (http://cbi.khu.ac.kr/RISD_DB.html) (Supplementary Table 1). The *mhz1-1* was used as male parent and crossed with *Osers2*, *Osetr2*, *Osein2*, and *OsEIN2-OE* transgenic lines to generate double mutants for genetic interaction analysis. The *Osers2*[d]/*MHZ1OE 4-4* line was derived from crossing *Osers2*[d] with *MHZ1OE 4-4* and the *Osers2*[d]/*MHZ1OE 3-5* line was generated by overexpressing *MHZ1* in *Osers2*[d] mutant through transgenic approach. To generate the *MHZ1-OE/Osein2* double-mutant, *MHZ1-OE* transgenic line was used as male parent and crossed with *Osein2*. F2 populations were used for genotyping and F3 or F4 populations were phenotypically and/or genotypically analyzed. The *mhz1-5* mutant with *Tos17* insertion was requested from the rice mutant database in Japan (https://pc7080.abr.affrc.go.jp/~miyao/pub/tos17/index.html.en). The *Osahp1*, *Osahp2* and *Osrr21* single mutants were generated through an CRISPR/Cas9 approach. SG sequences are as follows: *OsAHP1*, GTTGAGCTGGCTGGTCAGCG; OsAHP2, GATCTCGTTGAT-GATCCTGT; OsRR21, GGGACAGATATCGTTATGAA. To generate the *Osahp1 Osahp2* double-mutant, Nip rice was transformed with vectors carrying two SG sequences targeting *OsAHP1* and *OsAHP2*. After sequencing the genomic sequences of the two genes in 24 transgenic T0 lines, no *Osahp1 Osahp2* double-mutant was identified. An *Osahp1* (heterozygous)/*Osahp2* (homozygous) plant was self-crossed and the self-bred progenies (168 seeds) were germinated and grown under 10 ppm ethylene. After phenotype observation, seedlings were numbered and *OsAHP1* and *OsAHP2* genomic sequences were sequenced. Homozygous *Osahp1 Osahp2* plants were identified.

**Gene expression analysis by real-time PCR**. Two or three-day-old etiolated rice seedlings were treated with air or ethylene before roots and shoots were harvested for RNA extraction. Total RNA was extracted using TRIZOL reagent (Invitrogen). The complementary DNAs (cDNAs) were synthesized using cDNA Synthesis Kit (M-MLV Version) (TaKaRa) and then subjected to real-time PCR. Real-time PCR was conducted according to the instructions of TransStart Green qPCR SuperMix (TransGen Biotech, China). *OsActin2* was used for internal control. The primers are listed in Supplementary Table 1. The experiments were repeated independently for at least three times and the results were consistent. One set of results were shown.

**GUS Staining**. Tissues and organs were fixed in 90% acetone on ice for 15 min. After washing with staining buffer (100 mM $Na_3PO_4$ buffer pH 7.0, 10 mM EDTA, 5 mM potassium ferricyanide, 5 mM potassium ferrocyanide, 0.1% Triton X-100), the samples were soaked in staining solution (staining buffer containing 0.5 mg/mL X-Gluc (Sigma, B8049) for 10 min in a vacuum system. The samples were incubated at 37 °C in the dark. Green tissues were decolorized with 70% ethanol. The samples were observed using stereo microscopy (Leica, M165 FC).

**Proteins expression and phosphorylation assay**. The cDNA fragments encoding various protein versions were fused with *GST* in pGEX-6p-1 vector or maltose-binding protein (MBP) gene in pMAL-2C vector and proteins were expressed in BL21 (DE3) pLysS. GST-MHZ1(365–968 aa) containing kinase domain and receiver domain, and its mutant versions, GST-KD containing only the kinase domain (365–655 aa), and MBP-KD or its mutant versions were all expressed. Full-length MHZ1(1–968 aa) was not successfully expressed. Four OsHPts were expressed except OsPHP3 because the *OsPHP3* may be a pseudogene. The full-length *OsPHP1* (Os01g54050), *OsAHP1* (Os08g44350), *OsAHP2* (Os09g39400) and *OsPHP2* (Os05g09410) gene were fused with *GST* and similarly expressed. The fragments encoding the receiver domain of OsRR21 (1–129 aa, Os03g12350) or OsRR26 (1–123 aa, Os01g67770) were fused with *MBP* and expressed. Different truncated versions of ethylene receptors (ERS2ΔTM, ERS2GAF, ERS2HK, ERS1ΔTM, ETR2GAF + KD) were fused with GST and expressed. All genes encoding the mutant protein versions were generated by site-directed mutagenesis through overlapping extension PCR. The transfected *E. coli* cells were cultured at 25 °C and proteins were induced by addition of 0.8 mM IPTG. The proteins were purified with Glutathione sepharose for GST fusions or with Amylose Resin for MBP fusions.

Purified proteins were used for protein kinase assay according to our previous protocol[29]. Around 1–5 µg purified MHZ1 proteins, 5 µg purified OsHPts and/or OsRRs were used for autophosphorylation assay and/or phosphorelay analysis. Reactions were started by adding [γ-³²P]ATP, incubated at 25 °C for 45 min and then terminated by the addition of 5 × Loading buffer. Cation dependence of GST-MHZ1 kinase activity was tested at a physiological level of ATP (0.5 mM)[40]. To test the inhibition effect of ethylene receptors on MHZ1 kinase activity, 0.5–4 µg truncated OsERS2 or other ethylene receptor proteins were added to the reaction systems before adding [γ-³²P]ATP. Samples were subjected to sodium dodecyl sulfate–polyacrylamide gel electrophoresis (SDS-PAGE) using 10% polyacrylamide gels. After electrophoresis, the gel was dried for 4 h at 80 °C by the gel dryer and then the dry gel was exposed to X-ray films for autoradiography.

**Rice transformation and phenotype analysis**. The vector pCambia2300 was used for construction of complementation vector and overexpression vector. For complementation, the vector contained the *MHZ1* genomic sequence (5.623 kb), the 4.414 kb sequence upstream of *MHZ1* ATG (promoter) and the 1 kb sequence downstream of *MHZ1* TGA. This full-length sequence (11.037 kb) was cloned by adding three separated regions: 1–2249 bp region with Sse8387 I and *Bam*H I sites, the middle region (2249-8668 bp) with *Bam*H I and *Eco*R V, and the last region with *Eco*R V and *Sal* I. For overexpression, the *MHZ1*-coding sequence fused with sequence encoding FLAG tag was inserted into the pCambia2300 and the gene was driven by the *MHZ1* native promoter (3 kb). The 3 kb *MHZ1* native promoter was also fused with *GUS* gene in pCambia2300 for rice transformation and examination of promoter activity. Mutations of the G1 box (G588A, G590A) in the kinase domain of MHZ1, conserved His-containing site (H375Q) and conserved Asp (D824A) in receiver domain were generated in *MHZ1* cDNA by site-directed mutagenesis using overlapping extension PCR. To construct the *OsRR21* over-expression vector, open reading frame of the gene with restriction sites was amplified by PCR and cloned into cloning vector pEASY-Blunt (TRANSGENE BIOTECH), and subsequently cloned into binary vector pCambia2300-35S-OCS at the sites of *Bam*H I/*Sal* I. Plasmids were transfected into agrobacterium EHA105 and further subjected to rice transformation following previous protocol[29].

**OsEIL1 binding and activation of the *MHZ1* promoter**. GST-N-terminal OsEIL1 (amino acids 1–350) recombinant protein was expressed and purified according to our previous description[34]. Single-stranded complementary oligonucleotide fragments harboring the EBS elements (222 bp and 452 bp upstream of *MHZ1* start codon) were synthesized (Invitrogen) and labeled by biotin using the Biotin 3′-end DNA-labeling Kit (Thermo Fisher Scientific). Following assay was carried out according to manufacturer's protocol (LightShift Chemiluminescent EMSA Kit; Thermo Fisher Scientific).

For testing of the transactivation of the *MHZ1* promoter activity in tobacco leaves, the open reading frame of *OsEIL1* was cloned into pCambia2300-35S-OCS to generate *Pro35S:OsEIL1* vector. *MHZ1* promoter (2.16 kb DNA sequence upstream of ATG) was cloned to drive luciferase (LUC) gene expression. A combination of vectors containing *Pro35S:OsEIL1* and *ProMHZ1:LUC* were cotransformed into *A. tumefaciens* stain EHA105 and subsequently infiltrated into young leaves of *Nicotiana tabacum*. Plants were grown for 2.5 days with 16 h of light/8 h of dark at 24 °C before charge-coupled device (CCD) imaging. LUC activity was observed with a low-light cooled CCD imaging apparatus (iXon; Andor Technology). A combination of vector control and vector containing *ProMHZ1:LUC* was used as negative control.

**Membrane-based Y2H assay**. Coding sequence of OsERS2 was cloned into the bait vector pBT3-STE (OsERS2-Cub) and MHZ1 into the prey vector pPR3-N (NubG-MHZ1) from the DUAL membrane starter kit SUC (Dualsystem Biotech). The bait and prey constructs were then cotransformed into the yeast strain NMY51. Positive transformants were selected on SD-Trp-Leu medium, and protein–protein interactions were detected on SD-Trp-Leu-His-Ade medium. Combination of pTSU2-APP and pNubG-Fe65 (provided in the kit) was used as a positive control. Combinations of NubG-MHZ1 and pTSU2-APP, OsERS2-Cub and pNubG-Fe65 were used as negative controls.

**Pull-down of MBP-MHZ1 with GST-ERS2ΔTM**. To carry out the GST pull-down assay, the coding sequences of *MHZ1* and *OsERS2ΔTM* were cloned into the *pMAL-c2* and *pGEX-6P-1* plasmids, respectively, to make *MBP-MHZ1* and *GST-ERS2ΔTM* constructs. The constructs were then transformed into BL21(DE3). The transfected *E. coli* cells were cultured at 25 °C and proteins were induced by addition of 0.2 mM IPTG. After sonication and centrifugation, lysate supernatants containing GST-ERS2ΔTM recombinant protein were incubated with Glutathione sepharoses at 4 °C for 2 h. The sepharoses were washed with phosphate-buffered saline (PBS; pH 7.3) for three times before being incubated with supernatants containing MBP-MHZ1 recombinant protein at 4 °C for 2 h. The sepharoses were washed with PBS (pH 7.3) for three times. Harvested beads were boiled with 2 × SDS loading buffer before running the SDS-PAGE and immunoblotted with anti-GST, anti-MBP antibodies.

**Co-IP assays**. For coimmunoprecipitation of MHZ1 with ERS2ΔTM, constructs containing MHZ1-FLAG and ERS2ΔTM-GFP were cotransformed into rice protoplasts. Total proteins were extracted by homogenizing the protoplasts in 0.5 mL IP buffer (50 mM HEPES pH 7.5, 150 mM NaCl, 0.5 mM EDTA, 0.5% NP-40, 50 µM MG132, 2% (v/v) protease inhibitor cocktail) and incubating the samples on ice for 15 min. The samples were centrifuged at 12,000 rpm for 10 min at 4 °C before the supernatants were incubated with 25 µL of GFP-Trap beads for 2 h at 4 °C. After being washed for three times with wash buffer (10 mM Tris-HCl pH 7.5, 150 mM NaCl, 0.5 mM EDTA), the beads were collected and resuspended with 50 µL 2 × SDS-PAGE loading buffer and heated at 95 °C for 5 min. The eluted immunoprecipitates were immunoblotted with anti-GFP, anti-FLAG, and anti-UGPase antibodies. For interactions of MHZ1 with OsERS1 and OsETR2, constructs containing MHZ1-FLAG and OsERS1ΔTM/OsETR2ΔTM-GFP were cotransformed into rice protoplasts. Total proteins were immunoprecipitated with GFP-Trap and immunoblotted with anti-GFP, anti-FLAG, and anti-Actin antibodies.

For interaction domain mapping of MHZ1 and OsERS2, constructs containing MHZ1-FLAG, ERS2KD/GAF-GFP were cotransformed into rice protoplasts. Total proteins were immunopreciptated with anti-GFP affinity gel and immunoblotted with anti-FLAG, anti-GFP antibodies.

**Membrane recruitment assay**. To perform the membrane recruitment assay, MHZ1-GFP and OsERS2-mCherry proteins were expressed separately or together in tobacco leaf epidermal cells and fluorescence was examined. The images were taken using a confocal microscopy (Zeiss LSM 710). Excitation/emission wavelengths were set at 488 nm/500-530 nm for GFP and 561 nm/582-639 nm for mCherry.

**Histidine phosphorylation state detection**. To detect the histidine phosphorylation state of MHZ1 in *Osers2^d* mutant, vector carrying MHZ1-FLAG was transformed into protoplasts of WT and *Osers2^d*. Total proteins were extracted and MHZ-FLAG protein was immunopreciptated with anti-FLAG affinity gel and immunoblotted with anti-FLAG or anti-1-p-His antibodies (Millipore, MABS1330).

**Total and membrane protein isolation**. To analyze the localization of MHZ1 protein, vector carrying MHZ1-FLAG was transformed into protoplasts of WT and *Osers2^d* mutant. Total and membrane proteins were isolated as described by Ma[33]. Equal amounts of total protein (T), soluble protein (S), and microsomal membranes (M) were immunoblotted for MHZ1, BIP (ER membrane marker), and UGPase (cytoplasm marker).

**Measurement of ethylene production**. To test the ethylene production of different mutants, seedlings were grown in 40-mL uncapped vials for 7 days in dark at 28 °C. The

vials were then sealed with rubber syringe caps for 17 h. One milliliter of headspace of each vial was measured by using gas chromatography (GC-2014; Shimadzu)[35].

**Screening for supperssors of OsEIN2 (SOE).** For generation of *SOE* lines, seeds of *OsEIN2* overexpression line *OsEIN2OE-44* were soaked in water for 16 h at room temperature. The seeds were then treated with 0.6% EMS (Sigma, M0880) for 8 h at room temperature. The EMS-treated seeds were germinated at 37 °C and grown in the field. M2 generation seeds of the EMS-mutagenized lines were used for mutant screening[31].

**Statistical analysis.** The relative root or coleoptile length of each mutant is analyzed relative to the length in untreated conditions. All of the data were analyzed using a one-way ANOVA (LSD *t*-test) for the test groups.

**RNA-seq analysis.** To carry out the RNA-seq analysis, etiolated seedlings of WT, *mhz1*, *Osein2/mhz7* and *Oseil1/mhz6* were grown in the dark at 28 °C for 2 days before treated with air or 10 ppm ethylene for 3 h. Roots of WT and different mutants were subjected to RNA-seq analysis with three biological replicates. The clean data was mapped to rice genome by TopHat and analyzed with Cufflinks software. In WT and different mutants, genes with at least twofold changes in ethylene compared with those in air are marked as ethylene-inducible genes [$\log_2$(fold change) $\geq 1$, *q*-value < 0.05] or ethylene-repressive genes [$\log_2$(fold change) $\leq -1$, *q*-value < 0.05] genes. In WT, ethylene inducible and repressible genes are defined as ethylene-response genes (ERGs). In *mhz1*, *Osein2* and *Oseil1* mutants, ERGs that no longer respond to ethylene [*q*-value $\geq 0.05$ or |$\log_2$(fold change)| < 1, *q*-value < 0.05], or exhibit an opposite ethylene response pattern compared with WT (induced by ethylene in WT, repressed by ethylene in mutants or repressed by ethylene in WT, induced by ethylene in WT) were identified as MHZ1-, OsEIN2-, or OsEIL1-dependent ERGs, respectively. The test status of each gene indicates whether it is calculated. Genes with a test status of "NOTEST" indicates that there are not enough alignments for testing.

**Locus identifiers.** The locus identifiers of genes referred to in this article is as follows: Os06g44410 (*MHZ1*), Os01g54050 (*OsPHP1*), Os08g44350 (*OsAHP1*), Os09g39400 (*OsAHP2*), Os05g09410 (*OsPHP2*), Os03g12350 (*OsRR21*), Os06g08400 (*OsRR22*), Os02g55320 (*OsRR23*), Os02g08500 (*OsRR24*), Os01g67770 (*OsRR26*), Os06g08340 (*OsERF002*), Os09g11480 (*OsERF063*), Os09g11460 (*OsERF073*), Os11g05740 (*OsRAP2.8*), Os07g26720 (*OsRRA5*), Os05g06320 (*OsERS2*), Os04g08740 (*OsETR2*), Os07g06130 (*OsEIN2*), Os03g20790 (*OsEIL1*).

**Reporting summary.** Further information on research design is available in the Nature Research Reporting Summary linked to this article.

## Data availability

The authors declare that all data supporting the findings of this study are available within the manuscript and the Supplementary Files or are available from the corresponding authors upon request. The source data underlying Figs. 1a, 1e, 1f, 2a-c, 3i, 4a-c, 4g-m, 5b, 5e and Supplementary Figs. 4,a, 4,d, 5, 7,a, 7,b, 7,e-g, 8,b, 9,a, 9,b, 10,c, 10,d, 11,a, 11,b, 14 and 15 are provided as a Source Data file. RNA sequence data reported in this paper have been deposited in the NCBI Sequence Read Archive (https://www.ncbi.nlm.nih.gov/sra) under BioProject accession PRJNA597369 and BioSample accessions (SAMN13673841–13673864). Data has also been deposited in the Genome Sequence Archive in BIG Data Center, Beijing Institute of Genomics (BIG), Chinese Academy of Sciences, under accession numbers CRA002197.

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

## Acknowledgements

This work is supported by the National Natural Science Foundation of China (31530004, 31670274, 31600980), the National Key Basic Research Project (2015CB755702), and the State Key Lab of Plant Genomics.

## Author contributions

J.S.Z., H.Z., K.X.D., B.M. and S.Y.C. designed the research; H.Z. and K.X.D. performed most of the research; B.M. isolated the mutants and performed some research. C.C.Y., C.Y. and H.C. helped analyze the ethylene response phenotype of the mutants. J.J.T., Y.H.H. and W.Q.C identified the SOE lines. Z.G.Z. and T.G.L. provided the original population for screening; Y.H. performed RNA-seq analysis; S.J.H. conducted some rice transformation; W.K.Z and X.Y.W contributed to material preparation and data analysis; J.S.Z., K.X.D., H.Z. and B.M. wrote the paper.

## Competing interests

The authors declare no competing interests.
