## [Peer Review File · Nature Communications]

Reviewers' comments:

Reviewer #1 (Remarks to the Author):

The authors are studying the role of the histidine kinase MHZ1/HK1 of rice. MHZ1 is a homologue of AHK5 in Arabidopsis and HK9 of maize. Unlike other His-kinases in these plants, MHZ1/HK1 and its homologues lack transmembrane domains, having an N-terminal extension followed by a His-kinase domain and a receiver domain. Prior work with AtAHK5 and ZmHK9 have provided evidence that they affect the ethylene response in roots and stomata, but these studies were fairly basic and do not provide a great deal of information about mechanism. The study here by Duan et al, confirms the expected role for MHZ1/HK1 as a positive regulator of ethylene-mediated root growth inhibition in rice, and extends upon this mechanistically to try to resolve how it interacts with the ethylene-signaling pathway. The authors do a nice job of genetically characterizing the role of MHZ1 in root growth, but the proposal that it does so through direct interaction with the receptors would still require additional experimentation to confirm.

Figs. 1-3 nicely establish MHZ1 as being required for ethylene's ability to inhibit root growth (Fig. 1), being an ethylene induced gene such that its expression would facilitate the ethylene response on root growth (Fig. 2), and that its histidine kinase activity is necessary for its role in regulating root growth (Fig. 3). The role for MHZ1 as a positive regulator for ethylene responses in the root is the opposite of what was found when examining AtAHK5, and is also a much stronger phenotype than what was observed in Arabidopsis, suggesting a much greater role for MHZ1/HK1 in the rice ethylene response.

It gets more complicated when trying to interpret how MHZ1 interacts with the ethylene pathway, although the authors have constructed a great number of mutant combinations in rice to get at this question. They find that the reduced root growth found when MHZ1 is overexpressed can be blocked by MCP or a dominant ethylene-insensitive version of the receptor ERS2, indicating that ethylene signaling is required for that response. They also find that overexpression of EIN2 can largely overcome the ethylene insensitivity of *mhz1*, indicating that a heightened ethylene response can circumvent the role that MHZ1 plays, even if this does not occur under normal conditions. RNA-Seq experiments demonstrate that the mutant *mhz1* affects the expression of a subset of ethylene-dependent genes, consistent with a role in regulating a subset of the ethylene response. This doesn't place MHZ1 at any particular point of interaction with the pathway, although it does indicate that there is overlap in the regulation of the ethylene root growth response between the standard ethylene signaling pathway and the MHZ1-mediated pathway.

The authors propose based on several biochemical experiments that MHZ1 interacts directly with the receptors. These Y2H, pulldown and co-IP experiments (Fig 4e-h) seem reasonable and suggest the possibility of interaction between the two types of His-kinases (MHZ1 and ERS2). This suggests that a complex could be formed between the cytoplasmic MHZ1 and the membrane localized ethylene receptors, allowing for cross-talk between the pathways. The authors also perform in vitro autophosphorylation assays of MHZ1 and find that the presence of ethylene receptor ERS2 can inhibit this autophosphorylation, suggesting that there could be a regulatory interaction between MHZ1 and ERS2. It's not clear from such in vitro experiments if effects on activity are secondary effects due to physical impedance of MHZ1 autophosphorylation and whether such an effect would occur in planta. Optimally, if there is regulation of MHZ1 by the ethylene receptors in planta, you would predict that this would be ethylene dependent, something that is not possible to show with the pull-down and in vitro kinase assays. The authors attempt an experiment along these lines (Fig. 4I), using an anti-His antibody but the results are not that conclusive, the antibody appearing to be fairly non-specific in terms of protein binding. From the data presented it appears that MHZ1 and the ethylene receptors could form a complex, but how regulation occurs within such a complex is still uncertain.

Major points:

1. The authors would like to propose a model in which the ethylene receptors directly regulate activity of MHZ1, but in planta support for this model is weak. Admittedly such experiments are not easy, but would be necessary to support the authors' model. Right now the authors have reasonable support for a physical interaction, and based on their in vitro phosphorylation data the possibility that an increase in the level of ethylene receptors could inhibit activity of MHZ1. Ethylene induces expression of ethylene receptors and so one would then predict that MHZ1 activity would decrease due to the increased abundance of receptors; however, this is opposite to the opposite model the authors propose. This consideration should be discussed.

It is also possible that ethylene regulates activity of the MHZ1 by inducing changes in activity of the ethylene receptors. This could arise due to changes in affinity for the MHZ1 or from effects on the phosphorylation of MHZ1. This is what the authors propose but the only experiment the authors present in support of this model appears to be flawed. In Fig. 4I, the authors compared His-phosphorylation of MHZ1, following transient expression in wild-type or *ers2-d* (ethylene insensitive mutant of receptor) protoplasts. For evaluating His-phosphorylation, they used an anti-His antibody, but from their blots this has a high degree of non-specific binding to proteins and they do not include the necessary controls to demonstrate that the immunodecorated bands are specific for His-phosphorylation. Specifically, many lower molecular weight bands of the MHZ1 IP are immunodecorated, including bands too small to contain the phospho-His—there is thus a high degree of artifactual binding of the antibody. An important control that would need to be included is to use the authors' MHZ1(H375Q) mutant, which would be unable to autophosphorylate and should therefore not be immunodecorated by the antibody.

In terms of their experimental design with the protoplast experiment, it also seems that the native level of ethylene receptors would not be sufficient to significantly affect autophosphorylation of the transient and highly expressed MHZ1, based on the stoichiometry needed in their in vitro experiments. The authors would probably need to also transiently overexpress the ethylene receptor.

The authors' model relies on co-localization of MHZ1 with the ethylene receptors, but this has not been demonstrated by the authors. The ethylene receptors have been found to be localized to the endoplasmic reticulum and golgi. MHZ1 is predicted to be a cytoplasmic protein, although its Arabidopsis homologue AtAHK5 is reported to localize in part to the plasma membrane (Desikan et al 2008), calling into question whether MHZ1 will be found associated with ethylene receptor under normal conditions. The authors could exploit their protoplast system to examine the interaction of the ethylene receptors with MHZ1 (whether they can localize it to the ER) as has been done in the past to examine the interaction of membrane-bound ethylene receptors with receptors lacking their transmembrane domains (Grefen et al 2008, Mol Plant, 1:308).

2. Improvements can be made to the presentation of the gene expression data (Fig 5F, TableS1).

(a) The cutoffs used to determine a lack of ethylene regulation are not defined. In addition, although most of the calls from the spreadsheet seem reasonable, there are clearly some errors in calling whether genes are regulated or not. For example LOC_Os01g38110.MSUv7.0 has a log fc for wt of 6.02, for *mhz1* of -2.29, but is referred to as 'no' for regulation by MHZ1. In other cases a decrease to negative log values are considered a 'yes' for regulation. This raises the question as to whether a consistent methodology is being used for making the calls.

(b) Please include data on actual expression levels, not just fold change.

(c) Many of the most highly induced ethylene-responsive genes are listed as #N/A for the EIN2 and EIL1 analysis. It is not clear why this is the case. Are these, in actuality, regulated genes that now have very low expression due to the lack of ethylene signaling, in which case they should be marked as Yes in terms of their regulation; this is where the analysis of actual expression level indicated above can provide additional information on regulation rather than just giving fold change.

(c) Please provide the complete set of ethylene-regulated genes for each mutant analysis, not just

the ones that are known to be ethylene-regulated in wt. i.e. do any genes show ethylene regulation in the mutants that do not do so in wt?

(d) Please include gene descriptions with the table, not just the gene ID.

In terms of interpreting the gene expression data, please provide a more detailed analysis of the ethylene and MHZ1-regulated genes, for example the effects on gene expression by MHZ1 that are well known to be induced by ethylene, and that are likely to be primary response genes. Provide GO analysis for the subset of MHZ1 regulated gene in comparison to the total ethylene regulated gene set. Does the subset of MHZ1-regulated genes suggest targets for the regulation of root growth?

3. The authors only really consider a single hypothesis to explain their results, but as indicated above their experiments are not conclusive as to where and how MHZ1 interacts with the ethylene pathway. It has been previously proposed that AtAHK5, the MHZ1 homologue of Arabidopsis, may be regulated by H₂O₂, with the ethylene crosstalk due to ethylene inducing the production of H₂O₂ (Desikan et al, 2008). This possibility is still consistent with the authors' data (i.e. that regulation of MHZ1 is due in part to a downstream product arising from ethylene signaling), and can explain the ability of MCP to reverse the root growth response of overexpressed MHZ1.

Minor Points:

1. Fig. 1g. Please set the relative expression scale such that wt-air is equal to 1, so that fold-change can be more accurately assessed.

2. Fig. 1g. These are an unusual set of genes by which to characterize ethylene-regulated gene expression. I assume that authors chose these based on their later RNA-seq analysis as some of those specifically regulated by MHZ1. It would be useful to include gene expression for some of the more common strongly-induced genes that are MHZ1 independent. Also see later comment about performing a more detailed analysis on the RNA-seq dataset.

3. Fig. 4k. Why is there no exposure shown for GST-MHZ1 without any additions of the competitor ERS2 versions?

4. Fig. 5g. I think it would be worthwhile to show the proposed signaling pathways in air and in ethylene, to make it clear that the authors are proposing that the ethylene receptors phosphorylate MHZ1 in the presence of ethylene. Such a model could be changed dependent on what the in planta studies on regulation show, as it has also been proposed that the receptors are active in air but not in ethylene.

Reviewer #2 (Remarks to the Author):

The author has isolated a new factor that mutation showed a strong ethylene insensitive phenotype in roots. They further provide extensive evidence to show that MHZ1 is a kinase that can phosphorylates AHP in vitro. They also provide genetics evidence showed that the conserved motifs are important for MHZ1 to function in ethylene mediated root growth inhibition. They also provide in vitro biochemistry data to show the conserved motifs are important for MHZ1 kinase activity.

Overall, the discovery in the manuscript is very interesting, the new factor could provide more insight in ethylene signaling in rice. There are some concerns:

Figure 1e and figure 1f, the same MHZ1OE lines displayed different phenotype in air?

According to the author proposed model, OsAHP1 or OsAHP2 should have ethylene responsive phenotype, which is a very important data to support their conclusion that should be provided in the manuscript.

In Figure 4a, the authors showed that *mhz1* rescues *ers2* mutant, which may indicate that MHZ1 is downstream of ERS2. However, in figure 4b, it clearly showed that *ers2* dominant gain of function mutant clearly represses MHZ1OE phenotype, which could indicate that ERS2 is downstream of MHZ1. Therefore, it is not conclusive that MHZ1 is downstream of ERS2 based on figure 4. Most likely they function in the complex?

Figure 5a, b and C clearly showed that MHZ1 is downstream of EIN2, because *mhz1* is partially rescued EIN2-OE phenotype.

Figure 5f provide strong molecular evidence that MHZ1 and EIN2 and EIL1 are in the same pathway. The model should be modified.

According to the model, *ahp* and *rr21* mutants should displayed similar phenotype as that of *mhz1*. The author should provide their mutant phenotypes in the response to ethylene.

Response to the reviewers' comments

Reviewer #1:

The authors are studying the role of the histidine kinase MHZ1/HK1 of rice. MHZ1 is a homologue of AHK5 in Arabidopsis and HK9 of maize. Unlike other His-kinases in these plants, MHZ1/HK1 and its homologues lack transmembrane domains, having an N-terminal extension followed by a His-kinase domain and a receiver domain. Prior work with AtAHK5 and ZmHK9 have provided evidence that they affect the ethylene response in roots and stomata, but these studies were fairly basic and do not provide a great deal of information about mechanism. The study here by Duan et al, confirms the expected role for MHZ1/HK1 as a positive regulator of ethylene-mediated root growth inhibition in rice, and extends upon this mechanistically to try to resolve how it interacts with the ethylene-signaling pathway. The authors do a nice job of genetically characterizing the role of MHZ1 in root growth, but the proposal that it does so through direct interaction with the receptors would still require additional experimentation to confirm.

【Response】 Thank you for your comments. Additional experiments have been carried out according to your suggestions to support the OsERS2-inhibited MHZ1-phosphorelay pathway. The added results and refined models have been integrated into our revised manuscript. Changes in our revised manuscript are described in the following detailed responses.

Figs. 1-3 nicely establish MHZ1 as being required for ethylene's ability to inhibit root growth (Fig. 1), being an ethylene induced gene such that its expression would facilitate the ethylene response on root growth (Fig. 2), and that its histidine kinase activity is necessary for its role in regulating root growth (Fig. 3). The role for MHZ1 as a positive regulator for ethylene responses in the root is the opposite of what was found when examining AtAHK5, and is also a much stronger phenotype than what was observed in Arabidopsis, suggesting a much greater role for MHZ1/HK1 in the rice ethylene response.

It gets more complicated when trying to interpret how MHZ1 interacts with the ethylene pathway, although the authors have constructed a great number of mutant combinations in rice to get at this question. They find that the reduced root growth found when MHZ1 is overexpressed can be blocked by MCP or a dominant ethylene-insensitive version of the receptor ERS2, indicating that ethylene signaling is required for that response. They also find that overexpression of EIN2 can largely overcome the ethylene insensitivity of *mhz1*, indicating that a heightened ethylene response can circumvent the role that MHZ1 plays, even if this does not occur under normal conditions. RNA-Seq experiments demonstrate that the mutant *mhz1* affects the expression of a subset of ethylene-dependent genes, consistent with a role in regulating a subset of the ethylene response. This doesn't place MHZ1 at any particular point of interaction with the pathway, although it does indicate that there is overlap in the regulation of the ethylene root growth response between the standard

ethylene signaling pathway and the MHZ1-mediated pathway.

【Response】 Thank you for your comments. From our genetic analysis, it is most likely that MHZ1 genetically acts at the ethylene receptor level (Fig. 4a, b, c). Physically MHZ1 interacts with OsERS2 and MHZ1 kinase activity can be inhibited by GAF domain of OsERS2 and other receptors (Fig. 4). However, the *MHZ1-OE* plants still have ethylene response in *Osein2* mutant background, suggesting that MHZ1 function is somehow partially independent of OsEIN2 function (Fig. 5d, e). We hence propose that the MHZ1-mediated pathway works together with the OsEIN2-mediated pathway to affect the same subset of genes to regulate root growth, as derived from the RNA-seq analysis (Fig. 5f, Supplementary Table 1). We further revised our model incorporating all your comments (Fig. 5g). This model may largely explain the genetic relation between MHZ1 and OsEIN2: While *mhz1* could partially suppress the *OsEIN2-OE* ethylene-response phenotype (Fig. 5a, b, c), *Osein2* also partially suppressed *MHZ1-OE* ethylene-response phenotype (Fig. 5d, e).

The authors propose based on several biochemical experiments that MHZ1 interacts directly with the receptors. These Y2H, pulldown and co-IP experiments (Fig 4e-h) seem reasonable and suggest the possibility of interaction between the two types of His-kinases (MHZ1 and ERS2). This suggests that a complex could be formed between the cytoplasmic MHZ1 and the membrane localized ethylene receptors, allowing for cross-talk between the pathways. The authors also perform in vitro autophosphorylation assays of MHZ1 and find that the presence of ethylene receptor ERS2 can inhibit this autophosphorylation, suggesting that there could be a regulatory interaction between MHZ1 and ERS2. It's not clear from such in vitro experiments if effects on activity are secondary effects due to physical impedence of MHZ1 autophosphorylation and whether such an effect would occur in planta. Optimally, if there is regulation of MHZ1 by the ethylene receptors in planta, you would predict that this would be ethylene dependent, something that is not possible to show with the pull-down and in vitro kinase assays. The authors attempt an experiment along these lines (Fig. 4l), using an anti-His antibody but the results are not that conclusive, the antibody appearing to be fairly non-specific in terms of protein binding. From the data presented it appears that MHZ1 and the ethylene receptors could form a complex, but how regulation occurs within such a complex is still uncertain.

【Response】 Thank you for your comments. We further investigated the relationship between ethylene receptors and MHZ1 in the revised manuscript. Added results include: 1. OsERS2 facilitated the ER membrane localization of MHZ1 in our membrane recruitment assay (Fig. 4h); 2. Transiently overexpressed OsERS2 and *Osers2^d* (receptor gain-of-function) proteins significantly suppressed MHZ1 phosphorylation in our protoplast expression system (Fig. 4m); and *Osers2^d* has a stronger interaction with MHZ1 than wild-type OsERS2 does (Supplementary Fig. 9a, b, c), and also exhibits a stronger inhibition effect on MHZ1 phosphorylation (Fig. 4m). These results have been integrated into our revised manuscript at result section.

Discussion part is also revised accordingly.

Major points:

1. The authors would like to propose a model in which the ethylene receptors directly regulate activity of MHZ1, but in planta support for this model is weak. Admittedly such experiments are not easy, but would be necessary to support the authors' model. Right now the authors have reasonable support for a physical interaction, and based on their in vitro phosphorylation data the possibility that an increase in the level of ethylene receptors could inhibit activity of MHZ1. Ethylene induces expression of ethylene receptors and so one would then predict that MHZ1 activity would decrease due to the increased abundance of receptors; however, this is opposite to the opposite model the authors propose. This consideration should be discussed.

【 Response 】 Thank you for your comments. In the revised manuscript, we demonstrated that the constitutively activated OsERS2^d has a stronger interaction with MHZ1 than wild-type OsERS2 does (Supplementary Fig. 9a, b, c), and exhibits a stronger inhibition effect on MHZ1 phosphorylation (Fig. 4m), indicating that the conformation changes of OsERS2 affect its inhibition effect on MHZ1 activity. We propose that upon ethylene binding, the conformation change of OsERS2 impaired its inhibition effect on MHZ1, and MHZ1-mediated phosphorelay is triggered (Please see our refined model in Fig. 5g). The ethylene-induced receptor gene expression may occur at a relatively later stage, and may function as a desensitizing approach for ethylene response in order for ethylene receptor to re-lock MHZ1 possibly in a non-phosphorylation state after the initial biochemical triggering of the signaling. The discussions have been integrated into the revised manuscript (Page 20, Line 429-433).

It is also possible that ethylene regulates activity of the MHZ1 by inducing changes in activity of the ethylene receptors. This could arise due to changes in affinity for the MHZ1 or from effects on the phosphorylation of MHZ1. This is what the authors propose but the only experiment the authors present in support of this model appears to be flawed. In Fig. 4l, the authors compared His-phosphorylation of MHZ1, following transient expression in wild-type or *ers2-d* (ethylene insensitive mutant of receptor) protoplasts. For evaluating His-phosphorylation, they used an anti-His antibody, but from their blots this has a high degree of non-specific binding to proteins and they do not include the necessary controls to demonstrate that the immunodecorated bands are specific for His-phosphorylation. Specifically, many lower molecular weight bands of the MHZ1 IP are immunodecorated, including bands too small to contain the phospho-His—there is thus a high degree of artifactual binding of the antibody. An important control that would need to be included is to use the authors' MHZ1(H375Q) mutant, which would be unable to autophosphorylate and should therefore not be immunodecorated by the antibody.

In terms of their experimental design with the protoplast experiment, it also seems that the native level of ethylene receptors would not be sufficient to significantly affect autophosphorylation of the transient and highly expressed MHZ1, based on the

stoichiometry needed in their *in vitro* experiments. The authors would probably need to also transiently overexpress the ethylene receptor.

【Response】 Thank you for your great and wonderful suggestion. We further conducted the *in vivo* kinase assay accordingly and added MHZ1(H375Q) mutant as a negative control (Fig. 4m). Compared with the control, no phosphorylated MHZ1 band at 100 kDa was detected in *mhz1*:MHZ1(H375Q)-FLAG protoplasts with anti-His antibody, suggesting that the immunodecorated bands at 100 kDa are specific for MHZ1 His-phosphorylation (Fig. 4m). Based on this, we found that overexpressed OsERS2 and Osers2^d proteins significantly suppressed MHZ1 phosphorylation in our protoplast system (Fig. 4m). And Osers2^d has an apparently much stronger inhibition effect on MHZ1 phosphorylation compared with wild-type OsERS2 (Fig. 4m), indicating that the activity of OsERS2 affects its inhibitory effect on MHZ1. The original *in vivo* kinase assay (Fig. 4l in the original manuscript) showing weak MHZ1 His-phosphorylation in Osers2^d background was moved to Supplemental Fig. 9d.

The authors' model relies on co-localization of MHZ1 with the ethylene receptors, but this has not been demonstrated by the authors. The ethylene receptors have been found to be localized to the endoplasmic reticulum and golgi. MHZ1 is predicted to be a cytoplasmic protein, although its Arabidopsis homologue AtAHK5 is reported to localize in part to the plasma membrane (Desikan et al 2008), calling into question whether MHZ1 will be found associated with ethylene receptor under normal conditions. The authors could exploit their protoplast system to examine the interaction of the ethylene receptors with MHZ1 (whether they can localize it to the ER) as has been done in the past to examine the interaction of membrane-bound ethylene receptors with receptors lacking their transmembrane domains (Grefen et al 2008, Mol Plant, 1:308).

【Response】 Thank you for your suggestion. We performed the membrane recruitment assay in tobacco leaf epidermal cells as we found that the fluorescence signal in rice protoplasts transformed with OsERS2-mCherry is very weak. Results show that MHZ1-GFP was mainly detected in the cytoplasm when expressed alone (Fig. 4h). When expressed together, MHZ1-GFP was found to co-localize with OsERS2-mCherry to the ER membrane (Fig. 4h), suggesting that OsERS2 facilitated the ER membrane localization of MHZ1. In addition, protein fractionation assay shows that quite amounts of MHZ1 protein were detected in the membrane fractions especially in the presence of Osers2^d, further supporting the association of MHZ1 with membrane-bound OsERS2 protein (Supplementary Fig. 9a).

2. Improvements can be made to the presentation of the gene expression data (Fig 5F, TableS1).

(a) The cutoffs used to determine a lack of ethylene regulation are not defined. In addition, although most of the calls from the spreadsheet seem reasonable, there are clearly some errors in calling whether genes are regulated or not. For example,

LOC_Os01g38110.MSUv7.0 has a log fc for wt of 6.02, for mhz1 of -2.29, but is referred to as 'no' for regulation by MHZ1. In other cases, a decrease to negative log values are considered a 'yes' for regulation. This raises the question as to whether a consistent methodology is being used for making the calls.

【Response】 Thank you for your comments. To fix the problems in our RNA-seq analysis, we reanalyzed the RNA-seq data and made a clear definition of the cutoffs in the revised manuscript (Methods, Line 694-709). In WT and different mutants, genes with at least two-fold changes in ethylene compared with those in air are marked as ethylene-inducible genes [$\log_2(\text{fold change}) \geq 1$, q-value < 0.05] or ethylene-repressible genes [$\log_2(\text{fold change}) \leq -1$, q-value < 0.05] genes. In WT, ethylene inducible and repressible genes are defined as ethylene-responsive genes (ERGs). In *mhz1*, *Osein2* and *Oseil1* mutants, ERGs that no longer respond to ethylene [q-value ≥ 0.05 or $\log_2(\text{fold change}) < 1$, q-value < 0.05], or exhibit an opposite ethylene response pattern compared with WT (induced by ethylene in WT, repressed by ethylene in mutants or repressed by ethylene in WT, induced by ethylene in WT) were identified as MHZ1-, OsEIN2- or OsEIL1-dependent ERGs, respectively. The test status of each gene indicates whether it is calculated. Genes with a test status of 'NOTEST' indicates that there are not enough alignments for testing. The newly obtained data are presented in Fig. 5i, and Supplemental Table 1. The corresponding text and discussion parts were also revised accordingly.

(b) Please include data on actual expression levels, not just fold change.

【Response】 Thank you for your suggestion. The original RNA-seq analysis was performed according to a method which presents the relative fold changes instead of gene expression levels (Pertea *et al*, 2016, Nat. Protoc.). To get the actual expression levels, we reanalyzed the RNA-seq data. Clean data was mapped to rice genome by TopHat and analyzed with Cufflinks software. The reanalyzed RNA-seq data was supplied in Supplementary Table 1 in our revised manuscript, including both actual expression levels (FPKM) and fold changes.

(c) Many of the most highly induced ethylene-responsive genes are listed as #N/A for the EIN2 and EIL1 analysis. It is not clear why this is the case. Are these, in actuality, regulated genes that now have very low expression due to the lack of ethylene signaling, in which case they should be marked as Yes in terms of their regulation; this is where the analysis of actual expression level indicated above can provide additional information on regulation rather than just giving fold change.

【Response】 Thank you for your comments. In our original RNA-seq analysis, samples with low reads number that cannot be calculated were eliminated from the calculation step, and a #N/A is presented. Based on your suggestion, we reanalyzed the RNA-seq data. Genes are now divided into two groups according to their test status (See Supplementary Table 1). Both actual expression levels and fold changes

are included for clarity.

(c) Please provide the complete set of ethylene-regulated genes for each mutant analysis, not just the ones that are known to be ethylene-regulated in wt. i.e. do any genes show ethylene regulation in the mutants that do not do so in wt?

【Response】 Thank you for your suggestion. We provided the complete set of ethylene regulated genes in the revised manuscript (Supplementary Table 1). A number of genes were found to be unaffected by ethylene in WT, while respond to ethylene in the ethylene insensitive mutants (*mhz1*, 532 genes; *Osei1*, 224 genes; *Osei2*, 88 genes). These genes are supposed not to participate in the ethylene response of WT rice under normal conditions, but somehow are abnormally activated or suppressed in the mutants, which is probably due to feedback regulations.

(d) Please include gene descriptions with the table, not just the gene ID.

In terms of interpreting the gene expression data, please provide a more detailed analysis of the ethylene and MHZ1-regulated genes, for example the effects on gene expression by MHZ1 that are well known to be induced by ethylene, and that are likely to be primary response genes. Provide GO analysis for the subset of MHZ1 regulated gene in comparison to the total ethylene regulated gene set. Does the subset of MHZ1-regulated genes suggest targets for the regulation of root growth?

【Response】 Thank you for your suggestion. We have added gene annotations to the genes from the RNA-seq data in our revised manuscript (Supplementary Table 1). A GO analysis for the subset of MHZ1-dependent ERGs in comparison to total ERGs was performed (Supplementary Fig. 11c). Interestingly, we found that the MHZ1-dependent ERGs are enriched in auxin signaling pathway and responses to different stimuli (Supplementary Fig. 11c), indicating that MHZ1 is likely to be involved in the crosstalk between ethylene, auxin and different stimuli to regulate root growth. For total ERGs, the regulation process of gene expression, as well as auxin response pathway are enriched (Supplementary Fig. 11c). Description of the result was integrated to the revised manuscript (Page 18, Line 357-379).

3. The authors only really consider a single hypothesis to explain their results, but as indicated above their experiments are not conclusive as to where and how MHZ1 interacts with the ethylene pathway. It has been previously proposed that AtAHK5, the MHZ1 homologue of Arabidopsis, may be regulated by H₂O₂, with the ethylene crosstalk due to ethylene inducing the production of H₂O₂ (Desikan et al, 2008). This possibility is still consistent with the authors' data (i.e. that regulation of MHZ1 is due in part to a downstream product arising from ethylene signaling), and can explain the ability of MCP to reverse the root growth response of overexpressed MHZ1.

【Response】 Thank you for your comments. To investigate the relationship between MHZ1 and H₂O₂, we treated *mhz1* with H₂O₂. Results showed that root of *mhz1* is

slightly insensitive to H₂O₂ (Supplementary Fig. 15), suggesting that MHZ1 may be involved in H₂O₂-regulated root growth. This result is consistent with the GO analysis that MHZ1-dependent ERGs are enriched in auxin signaling pathway and also responses to different stimuli. We propose that MHZ1 is likely involved in the crosstalk between ethylene, auxin and H₂O₂ to regulate root growth. We refined the working model (Fig. 5g) and added discussion to the revised manuscript (Page 23, Line 482-491).

Minor Points:

1. Fig. 1g. Please set the relative expression scale such that wt-air is equal to 1, so that fold-change can be more accurately assessed.

【Response】 Modified as suggested, thank you.

2. Fig. 1g. These are an unusual set of genes by which to characterize ethylene-regulated gene expression. I assume that authors chose these based on their later RNA-seq analysis as some of those specifically regulated by MHZ1. It would be useful to include gene expression for some of the more common strongly-induced genes that are MHZ1 independent. Also see later comment about performing a more detailed analysis on the RNA-seq dataset.

【Response】 Thank you for your suggestion. The ethylene responsive genes used in our qPCR analysis (*OsERF002*, *OsRAP2.8* and *OsRRA5*) were identified in our previous studies (Ma *et al*, 2013; Ma *et al*, 2014; Yang *et al*, 2015; Yin *et al*, 2015). In the revised manuscript, we compared the ethylene responsiveness of *OsERF002*, *OsRAP2.8*, *OsRRA5* with *OsERF063* and *OsERF073*, which were also previously identified ethylene responsive genes. Whereas the ethylene induction of all five genes were abolished or hampered in *Osein2*, only *OsRRA5*, *OsRAP2.8*, and *OsERF002* expression was affected by *mhz1* (Supplementary Fig. 11a, b), suggesting that the ethylene responsiveness of *mhz1* and *Osein2* is differential in terms of gene expression. The results have been integrated to the revised manuscript (Page 17, Line 364-370).

3. Fig. 4k. Why is there no exposure shown for GST-MHZ1 without any additions of the competitor ERS2 versions?

【Response】 Thank you for your comments. We reperformed the assay and GST-MHZ1 was added to the assay as a control group (revised manuscript, Fig. 4i).

4. Fig. 5g. I think it would be worthwhile to show the proposed signaling pathways in air and in ethylene, to make it clear that the authors are proposing that the ethylene receptors phosphorylate MHZ1 in the presence of ethylene. Such a model could be changed dependent on what the in planta studies on regulation show, as it has also been proposed that the receptors are active in air but not in ethylene.

【Response】 Thank you for your suggestions. Based on the findings in our revised manuscript, we modified our model (Fig. 5g): In the absence of ethylene, the ethylene receptors are in active conformations, which facilitates their interaction with MHZ1 and MHZ1 kinase activity is suppressed. Upon ethylene perception, the conformation change of the receptors may impair their interaction with MHZ1, releasing their inhibition on MHZ1, and triggering MHZ1-mediated phosphorelay for regulation of root growth. The modified model is shown in Fig. 5g. The H₂O₂ was also incorporated into the model.

Reviewer #2 (Remarks to the Author):

The author has isolated a new factor that mutation showed a strong ethylene insensitive phenotype in roots. They further provide extensive evidence to show that MHZ1 is a kinase that can phosphorylates AHP in vitro. They also provide genetics evidence showed that the conserved motifs are important for MHZ1 to function in ethylene mediated root growth inhibition. They also provide in vitro biochemistry data to show the conserved motifs are important for MHZ1 kinase activity. Overall, the discovery in the manuscript is very interesting, the new factor could provide more insight in ethylene signaling in rice. There are some concerns:

Figure 1e and figure 1f, the same MHZ1OE lines displayed different phenotype in air?

【Response】 Thank you for your comments. In our revised manuscript, we re-performed the two experiments simultaneously. Seedlings treated with air, ethylene or 1-MCP were grown under the same conditions for same time lengths (Fig. 1e). The possible difference between batches of experiments was reduced.

According to the author proposed model, OsAHP1 or OsAHP2 should have ethylene responsive phenotype, which is a very important data to support their conclusion that should be provided in the manuscript.

【Response】 Thank you for your suggestions. Mutants of *OsAHP1*, *OsAHP2* were generated through CRISPR/Cas9 and their ethylene responses were examined (Supplemental Fig. 7a, b, c). *Osahp1* and *Osahp2* single mutants had similar ethylene responses with that of WT (Supplemental Fig. 7a), while *Osahp1 Osahp2* double mutant exhibited ethylene insensitive root growth (Supplemental Fig. 7b, c), suggesting that OsAHP1 and OsAHP2 may play redundant roles in ethylene signal transduction. Description of the result was integrated to the revised manuscript (Page 11, Line 227-242). It should be mentioned that only limited number of double mutant seedlings were segregated from the self-crossed population of *Osahp1*(heterozygous)

Osahp2(homozygous) plant, probably due to the embryo development defect of the double mutant. This has been incorporated into the discussion part (Page 21, Line 455-459).

In Figure 4a, the authors showed that *mhz1* rescues *ers2* mutant, which may indicate that MHZ1 is downstream of ERS2. However, in figure 4b, it clearly showed that *ers2* dominant gain of function mutant clearly represses MHZ1OE phenotype, which could indicate that ERS2 is downstream of MHZ1. Therefore, it is not conclusive that MHZ1 is downstream of ERS2 based on figure 4. Most likely they function in the complex?

【Response】 Thank you for your comments. The genetic analysis of *mhz1* with *Oers2* and *Oers2^d* indicates that MHZ1 may function at the ethylene receptor level. Combining with the findings that OsERS2 physically interacts with MHZ1 (Fig. 4e, f, g) and inhibits MHZ1 kinase activity (Fig. 4j, k, l, m), we propose that MHZ1 may form a complex with OsERS2 and works under the direct regulation of OsERS2 to modulate root growth. The corresponding description was also revised (Page 12, Line 252-255; Page 13, Line 266-268).

Figure 5a, b and C clearly showed that MHZ1 is downstream of EIN2, because *mhz1* is partially rescued EIN2-OE phenotype.

【Response】 Thank you for your comments. While *mhz1* could partially suppress the *OsEIN2-OE* phenotype (Fig. 5a, b, c), *Osein2* also partially suppressed *MHZ1-OE* phenotype (Fig. 5d, e). The complexity of the genetic relationship between MHZ1 and OsEIN2 may be due to the fact that, while the MHZ1-mediated phosphorelay pathway is transcriptionally downstream of the OsEIN2-OsEIL1 pathway (Fig. 2a, b, c), it physically interacts with and inhibited by ethylene receptors. The MHZ1-mediated pathway may work together with the OsEIN2-OsEIL1 pathway to regulate root growth. We have modified the working model in Fig. 5g in the revised manuscript to reflect such a complexity.

Figure 5f provide strong molecular evidence that MHZ1 and EIN2 and EIL1 are in the same pathway. The model should be modified.

【Response】 Thank you for your suggestions. As you suggested, Fig. 5f is consistent with the fact that MHZ1 is transcriptionally downstream of OsEIN2 and OsEIL1 (Fig. 2). Meanwhile, as MHZ1 genetically acts at the ethylene receptors (Fig. 4a, b, c) and interacts with OsERS2, we propose that the MHZ1-OsAHP1/2-OsRR21 pathway works together with the OsEIN2-OsEIL1 pathway to modulate a subset of genes to regulate root growth. This model may explain the genetic relation between MHZ1 and OsEIN2: While *mhz1* could partially rescue the *OsEIN2-OE* phenotype (Fig. 5a, b, c), *Osein2* also partially suppressed *MHZ1-OE* phenotype (Fig. 5d, e). We refined our model accordingly in our revised manuscript (Fig. 5g) and try to reflect the possible relationship.

According to the model, *ahp* and *rr21* mutants should displayed similar phenotype as that of *mhz1*. The author should provide their mutant phenotypes in the response to ethylene.

【Response】 Thank you for your suggestions. Mutants of *OsAHP1*, *OsAHP2* and *OsRR21* were generated through CRISPR/Cas9 and their ethylene responses were investigated (Supplemental Fig. 7a, b, c, d). While *Osahp1* and *Osahp2* single mutants had similar ethylene responses with that of WT (Supplemental Fig. 7a), *Osahp1 Osahp2* double mutant exhibited ethylene insensitive root growth (Supplemental Fig. 7b, c), suggesting that *OsAHP1* and *OsAHP2* may play redundant roles in *MHZ1*-mediated ethylene signal transduction. Two *Osrr21* mutants respond normally to ethylene (Supplemental Fig. 7d). But transgenic lines overexpressing *OsRR21* (we transformed WT rice with Pro35s:OsRR21 vector and three high expression lines were tested) exhibited shorter roots compared with WT both in air and in ethylene (Supplemental Fig. 7d), and expression of ethylene-responsive genes are enhanced in these overexpression lines compared with WT (Supplemental Fig. 7e). Hence the response regulators may also play redundant roles in ethylene signaling. It should be noted that ethylene can induce expression of several other RR genes, and these genes may also contribute to the ethylene response (Supplementary Fig. 7g). Description of the result was integrated to the revised manuscript (Page 11, Line 227-248).

REVIEWERS' COMMENTS:

Reviewer #1 (Remarks to the Author):

The authors have added additional experimental data to support their model for the role of MHZ1/HK1 in ethylene inhibition of rice root growth. Some additional textual clarifications of their data that would improve the manuscript are indicated below.

1. Some aspects of the model should be clarified:

(a) The authors now provide an *in vivo*, protoplast-based assay demonstrating that overexpression of ERS2 can inhibit His-phosphorylation of MHZ1/HK1 (Fig. 4m). They do not as yet provide a clear demonstration that ethylene treatment affects such phosphorylation, which should be possible using the protoplast assay, and so need to make it clear that such ethylene-dependent regulation is still an hypothesis rather than a demonstrated fact. i.e for figure 5e, put a question mark above the line connecting inhibition of MHZ1 by the receptors in response to ethylene. For example, it is still possible that the interaction with the receptors serves a purely inhibitory role, and that it is an increase in free MHZ1 that regulates the downstream ethylene response.

(b) The model only shows co-regulated downstream genes co-regulated by MHZ1 and the ethylene pathway. It would clarify the model to also indicate that there are MHZ1-pathway independent genes regulated in the root by ethylene. This is supported by the RNA-seq analysis.

(c) Include unidentified type-B RR in the model, since their *rr21* mutant did not exhibit a ethylene mutant phenotype.

(d) Given the role of ethylene in stimulating expression of MHZ1 pathway elements, showing their text at a reduced size in the absence of ethylene would provide a point of clarification about signaling capacity.

2. In addition, unless there is a length limit, it would be useful for the authors to move some figure information from the supplement to the main body of the paper. This is based on the principle that data needed to support their model should be part of the main paper rather than the supplement.

(a) For Fig. 1f, include qRT-PCR analysis of ERF063 and ERF073 comparing their induction in root and shoot, as these are genes that based on the supplemental figure 11b, are ethylene/EIN2 regulated but MHZ1 independent.

(b) include data on CRISPR analysis of AHPs and type-B RRs, which form the basis for the authors' proposal that these function downstream of MHZ1 to regulate root growth in response to ethylene.

Other textual changes:

1. Title should include the name HK1, not just MHZ1, since that is the priority for naming of this gene and how it is more broadly known. Authors should reference prior papers that name gene OshK1.

2. The authors need to provide more background on HK1/MHZ1 in their Introduction and Discussion. Most relevant is the OshK1 paper from 2018 (Lehner et al, 2018, A histidine kinase gene is required for large radius root tip circumnutation and surface exploration in rice doi: <http://dx.doi.org/10.1101/437012>. bioRxiv). Lehner et al demonstrates a role for OshK1 in root growth and circumnutations, as well as providing some evidence for an OshK1 role in cytokinin signaling. On the other hand, prior work with Arabidopsis, indicates that ethylene stimulates nutations (Binder et al, 2006 Plant Phys 142:1690), which would fit with current MHZ1 study on the gene's role in root ethylene responses. This prior OshK1 root study should be introduced in the Introduction. The relationship of this study to the authors' findings should be incorporated into the Discussion.

3. The authors should give their *mhz* mutant alleles specific *hk* mutant designations in the text. These would presumably be as *hk1-4*, *hk1-5*, etc., since Lehner et al previously isolated *hk1-1*, *hk1-2*, and *hk1-3* in their study.

4. Line 122. The authors should state that MHZ1 is also homologous to ZmHK9, and state that these have been implicated in regulating the ethylene response.

5. Methods does not include information on how the CRISPRcas9 AHP and RR21 mutant alleles were generated. In addition, the authors indicate in their rebuttal letter that *ahp1/2* is embryo lethal, but do not indicate this in the manuscript—this should be added, as well as how it was then possible for them to obtain the *ahp1/2* mutant for *figS7b/c*.

6. Supplemental figure 7. Is data in 7e and 7g derived from qRT-PCR or from the RNA-seq experiment?

Reviewer #2 (Remarks to the Author):

The authors have done a significant revision with the manuscript, all my concerns were addressed. I have no further questions.

Response to the reviewers' comments

Reviewer #1:

The authors have added additional experimental data to support their model for the role of MHZ1/HK1 in ethylene inhibition of rice root growth. Some additional textual clarifications of their data that would improve the manuscript are indicated below.

1. Some aspects of the model should be clarified:

(a) The authors now provide an *in vivo*, protoplast-based assay demonstrating that overexpression of ERS2 can inhibit His-phosphorylation of MHZ1/HK1 (Fig. 4m). They do not as yet provide a clear demonstration that ethylene treatment affects such phosphorylation, which should be possible using the protoplast assay, and so need to make it clear that such ethylene-dependent regulation is still an hypothesis rather than a demonstrated fact. i.e for figure 5e, put a question mark above the line connecting inhibition of MHZ1 by the receptors in response to ethylene. For example, it is still possible that the interaction with the receptors serves a purely inhibitory role, and that it is an increase in free MHZ1 that regulates the downstream ethylene response.

【Response】 Thank you for your comments and suggestions. As you suggested, the ethylene-dependent regulation of MHZ1 activity by the ethylene receptors could be further explored while other possibilities cannot be excluded. We revised our model and added a question mark to this step following your suggestion.

(b) The model only shows co-regulated downstream genes co-regulated by MHZ1 and the ethylene pathway. It would clarify the model to also indicate that there are MHZ1-pathway independent genes regulated in the root by ethylene. This is supported by the RNA-seq analysis.

【Response】 Thank you for your suggestions. We have revised our model and MHZ1-independent genes are integrated into the model.

(c) Include unidentified type-B RR in the model, since their *rr21* mutant did not exhibit a ethylene mutant phenotype.

【Response】 Other OsRRs are added to our model as suggested, thank you.

(d) Given the role of ethylene in stimulating expression of MHZ1 pathway elements, showing their text at a reduced size in the absence of ethylene would provide a point of clarification about signaling capacity.

【Response】 We have revised the model according to your suggestions, thank you.

2. In addition, unless there is a length limit, it would be useful for the authors to move some figure information from the supplement to the main body of the paper. This is

based on the principle that data needed to support their model should be part of the main paper rather than the supplement.

(a) For Fig. 1f, include qRT-PCR analysis of ERF063 and ERF073 comparing their induction in root and shoot, as these are genes that based on the supplemental figure 11b, are ethylene/EIN2 regulated but MHZ1 independent.

【Response】 Thank you for your suggestions. The qPCR analysis of *ERF063* and *ERF073* has been integrated into Fig. 1f.

(b) include data on CRISPR analysis of AHPs and type-B RRs, which form the basis for the authors' proposal that these function downstream of MHZ1 to regulate root growth in response to ethylene.

【Response】 The phenotype analyses of *Osahp*, *Osrr21* mutants and *OsRR21* overexpression lines have been moved to Fig.3j and 3k as you suggested, thank you.

Other textual changes:

1. Title should include the name HK1, not just MHZ1, since that is the priority for naming of this gene and how it is more broadly known. Authors should reference prior papers that name gene OsHK1.

【Response】 Thank you for your suggestions. Title has been revised accordingly and the papers (Tsai *et al*, 2012; Lehner *et al*, 2018) was cited.

2. The authors need to provide more background on HK1/MHZ1 in their Introduction and Discussion. Most relevant is the OsHK1 paper from 2018 (Lehner *et al*, 2018, A histidine kinase gene is required for large radius root tip circumnutation and surface exploration in rice doi: <http://dx.doi.org/10.1101/437012>. bioRxiv). Lehner *et al* demonstrates a role for OsHK1 in root growth and circumnutations, as well as providing some evidence for an OsHK1 role in cytokinin signaling. On the other hand, prior work with Arabidopsis, indicates that ethylene stimulates nutations (Binder *et al*, 2006 *Plant Phys* 142:1690), which would fit with current MHZ1 study on the gene's role in root ethylene responses. This prior OsHK1 root study should be introduced in the Introduction. The relationship of this study to the authors' findings should be incorporated into the Discussion.

【Response】 Thank you for your suggestions. The description of the previous work on OsHK1 has been integrated into the Introduction part. Relationship between the previous studies on OsHK1 and our finding are also discussed in the Discussion section.

3. The authors should give their mhz mutant alleles specific hk mutant designations in the text. These would presumably be as hk1-4, hk1-5, etc., since Lehner *et al*

previously isolated hk1-1, hk1-2, and hk1-3 in their study.

【Response】 Revised as suggested, thank you.

4. Line 122. The authors should state that MHZ1 is also homologous to ZmHK9, and state that these have been implicated in regulating the ethylene response.

【Response】 Revised as suggested, thank you.

5. Methods does not include information on how the CRISPRcas9 AHP and RR21 mutant alleles were generated. In addition, the authors indicate in their rebuttal letter that *ahp1/2* is embryo lethal, but do not indicate this in the manuscript—this should be added, as well as how it was then possible for them to obtain the *ahp1/2* mutant for figS7b/c.

【Response】 Thank you for your comments. We have added the information on the generation of *ahp1*, *ahp2* and *rr21* mutants to the Method section (Materials, ethylene treatment and gene identification.). The possible defects of the *Osahp1 Osahp2* double mutant is discussed in the Discussion part.

6. Supplemental figure 7. Is data in 7e and 7g derived from qRT-PCR or from the RNA-seq experiment?

【Response】 Thank you for your comments. Data in Supplementary Fig. 7e, g is derived from qPCR analysis. We have revised the figure legend to make it clearly stated.

Reviewer #2 (Remarks to the Author):

The authors have done a significant revision with the manuscript; all my concerns were addressed. I have no further questions.